# Identification of a covert evolutionary pathway between two protein folds

Devlina Chakravarty[1], Shwetha Sreenivasan[2], Liskin Swint-Kruse [2] & Lauren L. Porter [1,3] ✉

Although homologous protein sequences are expected to adopt similar structures, some amino acid substitutions can interconvert α-helices and β-sheets. Such fold switching may have occurred over evolutionary history, but supporting evidence has been limited by the: (1) abundance and diversity of sequenced genes, (2) quantity of experimentally determined protein structures, and (3) assumptions underlying the statistical methods used to infer homology. Here, we overcome these barriers by applying multiple statistical methods to a family of ~600,000 bacterial response regulator proteins. We find that their homologous DNA-binding subunits assume divergent structures: helix-turn-helix versus α-helix + β-sheet (winged helix). Phylogenetic analyses, ancestral sequence reconstruction, and AlphaFold2 models indicate that amino acid substitutions facilitated a switch from helix-turn-helix into winged helix. This structural transformation likely expanded DNA-binding specificity. Our approach uncovers an evolutionary pathway between two protein folds and provides a methodology to identify secondary structure switching in other protein families.

Life is sustained by the chemical interactions and catalytic reactions of hundreds of millions of folded proteins. The structures and functions of these proteins are determined by their amino acid sequences[1]. As such, sequence changes have various functional effects, ranging from none to intermediate impairment to complete loss[2,3], with biological outcomes ranging from no observable effect to debilitating disease[4–6]. While many historical studies indicate that amino acid variation can locally or globally unfold protein structure[7,8], such changes typically do not remodel secondary structure, such as converting α-helices to β-sheets. These findings support the well-established observation that proteins with similar sequences have similar folds and execute similar functions. In turn, these similarities are used to classify protein folds into families[9–11] and underlie state-of-the-art protein structure prediction methods[12–14].

Nevertheless, recent work shows that a subset of amino acid changes can switch secondary structure. This process has been called "evolutionary metamorphosis[15]" and "evolved fold switching[16]".

For instance, the most frequent non-Hodgkin-lymphoma-associated mutation observed in human mycocyte enhancer factor 2 (MEF2) switches a C-terminal α-helix to a β-strand, likely impeding MEF2 function[17]. Furthermore, numerous single mutations deactivate the cyanobacterial circadian clock by preventing a transformation that is critical for its normal function – the switch of its C-terminal subdomain from a βααβ fold to an αββα fold[18]. Finally, for an engineered protein G variant, a single mutation or incorporation into a larger protein domain can switch the 3-α-helix bundle that binds human serum albumin to other folds with altered functions, such as an α/β-grasp fold that binds immunoglobulins or an α/β-plait ribosomal protein domain[19–23].

These examples suggest that evolved fold switching of secondary structures, via stepwise amino acid changes, may be one mechanism by which new protein folds originate in nature. If so, this evolutionary mechanism should be identifiable by searching for homologous protein sequences with different experimentally determined structures

[1]National Center for Biotechnology Information, National Library of Medicine, National Institutes of Health, Bethesda, MD 20894, USA. [2]Department of Biochemistry and Molecular Biology, The University of Kansas Medical Center, Kansas City, KS 66160, USA. [3]Biochemistry and Biophysics Center, National Heart, Lung, and Blood Institute, National Institutes of Health, Bethesda, MD 20892, USA. ✉e-mail: lauren.porter@nih.gov

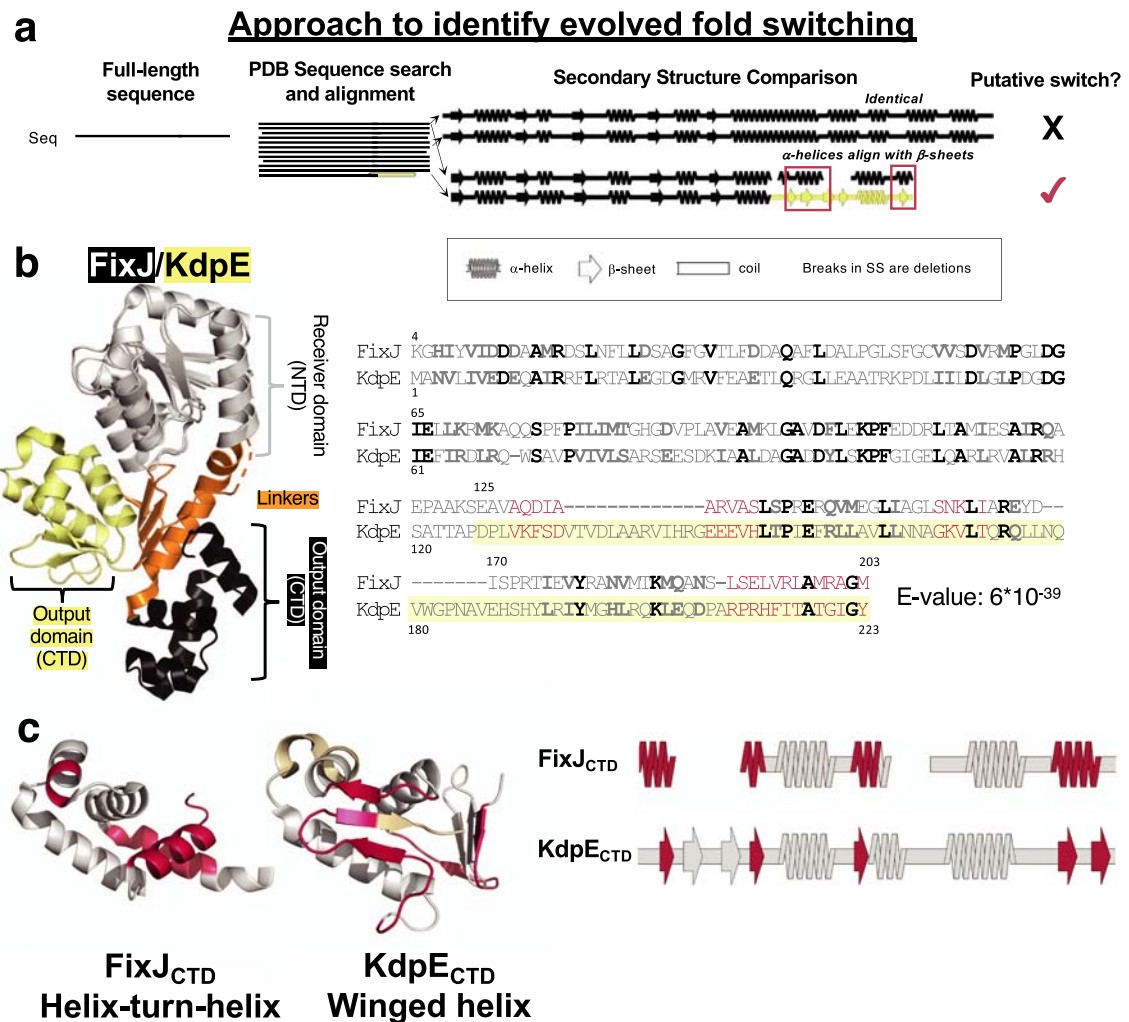

**Fig. 1 | A combined sequence-structure search indicates that mutations may have switched some secondary structures between tetrahelical helix-turn-helix (HTH₄) and winged helix (wH) proteins. a** Querying the full sequence of FixJ (HTH₄) against the PDB with one round of BLAST yielded a significant match with full-length KdpE (wH). Notably, in two regions, experimentally determined α-helices aligned with β-sheets. **b** A subsequent PSI-BLAST search confirmed a likely evolutionary relationship between the full-length FixJ and KdpE sequences; full-length structures are shown with conserved NTDs in gray, linkers in orange, HTH₄ CTD in black, and wH CTD in yellow. The resulting PSI-BLAST alignment includes the NTD and CTD (starting where KdpE sequence is highlighted in yellow); bold amino acids are identical (black) or similar (gray), regions were α-helices align with β-strands are pink; gaps are denoted '-'. **c** Regions of three-dimensional structure (left) and secondary structure (right) where PSI-BLAST aligns α-helices in the HTH₄ fold with sequences of β-strand in the wH fold (pink). Gray regions indicate conserved secondary and tertiary structure; beige regions in the wH correspond to its additional amino acids in the alignment, indicated as open spaces in the aligned secondary structure of FixJ (right). Source data are provided as a Source Data file.

(Fig. 1a). Similar approaches have successfully identified evolutionary relationships between protein fold families with conserved secondary structures but different tertiary arrangements[24,25].

However, observations of evolved secondary structure interconversion have been impeded by several technical barriers: (1) the limited abundance and diversity of sequenced genes, (2) the limited quantity of experimentally determined protein structures, and (3) the assumptions underlying the statistical methods used to infer homology. Indeed, all three limitations impacted the pioneering work of Cordes and colleagues, who identified a likely evolutionary relationship between the two distinctly folded transcription factors, P22 Cro and λ Cro[26–28]. Structurally, these two proteins share a 3-helical N-terminal core but have divergent C-terminal regions: P22 Cro's C-terminal region folds into two α-helices, whereas λ Cro's C-terminal assumes a β-hairpin. Although these differences could have arisen from evolved fold-switching, the data available were too limited to be conclusive: at the time of their study, the protein family comprised only 55 sequences and 5 solved structures (barriers (1) and (2)). The

authors also proposed the existence of barrier (3): since whole-database PSI-BLAST searches did not identify P22 Cro and λ Cro as homologous, the authors concluded that[27], "profile-based methods might be intrinsically ill suited…when wholesale structural change has occurred, since sequence conservation patterns will change in such a case."

Since the aforementioned study was performed nearly 20 years ago, the number of available sequences in the RefSeq[29] database has increased by three orders of magnitude, and the number of experimentally determined structures deposited in the Protein Data Bank (PDB) has increased by a factor of 7[30,31]. Thus, we hypothesized that sufficient protein sequence and structure information are now available to detect stepwise amino acid changes that lead to evolved fold switching.

To that end, we searched for evidence among a large family of bacterial response regulators comprising ~600,000 sequences and 76 unique, experimentally determined structures. Each homolog in this family constitutes one-half of a bacterial "two-component system"; the

other half is a cognate sensor protein[32]. These protein pairs work together to allow bacteria to respond to their environments through chemotaxis[33], antibiotic resistance[34], oxygen sensing[35], and more[36]. To carry out its function, each sensor protein has an extracellular domain that binds a triggering ligand, thereby activating the sensor's histidine kinase domain to phosphorylate its cognate response regulator at a conserved aspartate in the N-terminal receiver domain. In turn, this modification causes the response regulator's C-terminal "output" domain to mount the organism's response, such as altered transcription regulation[37].

Structurally, the response regulator proteins share a common N-terminal domain architecture, whereas structural differences among their C-terminal domains have been used to divide them into subfamilies[37,38]. Nearly 50% of the C-terminal domains fold into either helix-turn-helix (HTH) or winged helix (wH) DNA-binding domains[37]. (This ~50% corresponds to the ~600,000 sequences mentioned above). Both C-terminal domain folds comprise a core 3-helix bundle flanked by either (1) an N-terminal helical linker and a 4th C-terminal helix (e.g., a tetrahelical HTH, or $HTH_4$) or (2) a four-stranded N-terminal β-sheet (here called a linker for ease of comparison) and a C-terminal β-hairpin (or "wing", Fig. 1b and c). On average, response regulators with $HTH_4$ output domains are ~30 residues shorter than their wH counterparts.

Common evolutionary descent of the response regulator $HTH_4$ and wH domains was suggested previously[39]. However, an evolutionary mechanism could not be detected, again most likely due to the paucity of sequence and structure information available at the time of study. Thus, it has been unclear whether the differences in CTD secondary structures resulted from sequence insertions, complete or partial domain recombination, stepwise amino acid changes (e.g., evolved fold switching), or some combination of the three.

In this work, we report strong statistical support for evolved fold switching of C-terminal secondary structure in $HTH_4$ and wH domains and propose a putative evolutionary pathway between the two folds. First, we showed that the C-terminal α-helix of the $HTH_4$ shares an evolutionary relationship with the β-sheet wing of the wH (Figs. 1 and 2). This relationship was then reinforced through multiple statistical analyses of phylogenetic relationships, ancestral sequence reconstruction with AlphaFold2 models, and functional analyses. All lines of evidence consistently point to an evolutionary trajectory by which an α-helix transformed into a β-sheet through stepwise mutation(s). Our results suggest how stepwise mutations can switch protein secondary structure and provide methodology to identify evolved fold switching in other protein families.

## Results

### Apparent homology between bacterial response regulators with $HTH_4$ and wH CTDs

We previously used protein BLAST[40] to search the PDB for pairs of protein sequences with high sequence identity (≥70% though not identical) but divergent, experimentally determined secondary structures[41] (Fig. 1a). This study supports the hypothesis that homologous proteins can switch folds through stepwise mutation but could not provide a detailed description of how the structural transitions occurred. Indeed of the fold-switching proteins reported, NusG had the largest sequence set, with ~16,000 non-redundant sequences[42]; however, these sequences are unreliably annotated[42] and the fold transition/s is/are difficult to identify[43], confounding phylogenetic analyses that could potentially reveal the fold-switch transition.

Here, we reasoned that searching families with larger numbers of sequences would enhance the statistics underlying homology inference, boost fold annotation accuracy, and enable the statistically significant phylogenetic analyses required to identify homologous but distinctly folded proteins. Larger families may also afford the ability to identify evolved fold switching pathways among sequences with ≤70% identity. To that end, we used all ~150,000 sequences in the PDB to query all other sequences with divergent secondary structures ("Methods" section) and identified sequence matches with e-values of 1e-04 or lower. Lower e-values indicate that a match is increasingly unlikely to arise by chance, allowing homology to be inferred[44]. Our threshold of 1e-04 is conservative; 5e-02 is often used to infer homology[40] and some sequences with even higher e-values are also homologous[40].

Among the pairs of potential fold-switching homologs in the PDB, we identified a match between the full-length structures of FixJ from *Bradyrhizobium japonicum* (query) and KdpE from *Escherichia coli*, with an e-value of 1e-07. Importantly, $FixJ_{PDB}$ and $KdpE_{PDB}$ are defined as having different folds by several independent annotators, including Pfam, ECOD, and SCOP ("Methods" section). Both $FixJ_{PDB}$ and $KdpE_{PDB}$ are response regulators of bacterial two-component systems. These proteins are highly abundant within and among myriad bacterial species. Sequences for >1,000,000 diverse genes are present in the nr database, which is nearly 2 orders of magnitude larger than the NusG family mentioned before.

Structurally, the N-terminal domains (NTDs) of $FixJ_{PDB}$ and $KdpE_{PDB}$ showed high sequence and structural similarities (Fig. 1b, left), whereas their linkers and DNA-binding C-terminal domains (CTDs) showed modest sequence similarities and striking differences in secondary structure: $FixJ_{PDB}$'s CTD comprises a tetrahelical helix-turn-helix ($HTH_4$) architecture, whereas $KdpE_{PDB}$'s CTD comprises a winged helix (wH, Fig. 1). The $KdpE_{PDB}$ CTD is also 15 aa longer than that of $FixJ_{PDB}$. Nonetheless, FixJ's helical linker aligned partially with the four β-sheets of KdpE's CTD. (For ease of comparison, we call both regions, "linkers".) Furthermore, the C-terminal α-helix of $FixJ_{PDB}$ aligns with the C-terminal β-hairpin of $KdpE_{PDB}$'s CTD, also known as its "wing".

In contrast to queries with the full-length proteins, BLAST and PSI-BLAST searches of the PDB using the sequences of isolated CTDs from either $FixJ_{PDB}$ or $KdpE_{PDB}$ as queries only identified sequences from the same fold families ($HTH_4$ or wH). Sequences encoding the alternative structure were not identified.

Two possibilities could explain these conflicting results. First, in the full-length sequences, the strong similarities of the NTD could erroneously give rise to the CTD alignment through "homologous overextension", in which flanking, non-homologous sequences are erroneously included in a local sequence alignment[45]. In this case, the distinctly folded CTDs would *not* share a common ancestor. Instead, genes encoding the separate CTDs likely recombined with genes encoding the NTDs of response regulators. Consistent with this possibility, the alignment coverage after our initial BLAST search included only 52% of the CTD sequence. Alternatively, the $HTH_4$ and wH domains could share a common ancestor that is difficult to robustly infer from the isolated, divergent CTD sequences. In this case, searching with complete sequences (NTD + CTD) produced statistically significant alignments that correctly suggested an evolutionary relationship between alternatively folded CTDs. Indeed, the second phenomenon was proposed for both the Cro proteins[26-28] and bacterial NusG transcription factors[46].

To further discriminate whether our initial $FixJ_{PDB}/KdpE_{PDB}$ $HTH_4$/wH match indicated a true evolutionary relationship or resulted from faulty homologous overextension, we next used full-length $FixJ_{PDB}$ to query the PDB with 3 rounds of PSI-BLAST[40], an iterative algorithm that identifies conservation patterns among homologous protein sequences. Unlike the faster BLAST algorithm (which identifies matches using pairwise identities between the query sequence and entries in a sequence database), PSI-BLAST searches for sequences that match conservation patterns within a set of homologous sequences used to generate a position-specific scoring matrix. This matrix stores scores for substituting one amino acid for another in each sequence position and is updated after each PSI-BLAST iteration if new sequences are hit in the search. As such, PSI-BLAST identifies hidden conservation patterns characteristic to a given protein family

that cannot be detected by BLAST. Indeed, PSI-BLAST identified stronger conservation patterns between sequences encoding HTH₄ and wH folds. This alignment approach also shifted the alignment registers of the CTDs, so that 97% of the FixJ$_{PDB}$ sequence aligned with KdpE$_{PDB}$ with an *e*-value of $6 \times 10^{-39}$ (Fig. 1b, right). This result supports the hypothesis that the HTH₄ and wH folds of the FixJ and KdpE CTDs are distant homologs rather than alignment artifacts.

Furthermore, for 11 of the top 20 PSI-BLAST matches from this search, the CTDs assumed the same wH fold as KdpE$_{PDB}$, whereas the other 9 matches assumed the same HTH fold as the FixJ$_{PDB}$ query (Supplementary Table 1). A reciprocal, three-round PSI-BLAST search using the full-length KdpE$_{PDB}$ sequence as query aligned 90% of this protein with FixJ$_{PDB}$, with an e-value of $10^{-29}$. Notably, sequences of isolated DNA-binding domains with HTH folds were matched with the CTD of KdpE$_{PDB}$ (wH), and sequences of isolated DNA-binding domains with wH folds were matched with the sequence of FixJ$_{PDB}$'s CTD (HTH₄, Supplementary Table 2). Together, these results indicate that: (1) HTH₄ and wH domains share a common ancestor[39] and (2) the use of full-length sequences in our analyses, rather than isolated domains, is both legitimate and necessary to identify the relationship. Thus, all subsequent searches used full-length sequences as queries, unless otherwise noted.

Further examination of the aligned FixJ$_{PDB}$ HTH₄ and KdpE$_{PDB}$ wH folds revealed regions of structural similarity and dissimilarity: both folds share a conserved trihelical core[39] (Fig. 1c). By contrast, striking regions of dissimilarity are evident between (1) FixJ$_{PDB}$'s α-helical interdomain linker and KdpE's corresponding quadruple-stranded β-sheet; long gaps in this alignment suggest that KdpE$_{PDB}$'s linker region was extended through an insertion, and (2) FixJ$_{PDB}$'s C-terminal helix aligned with KdpE$_{PDB}$'s C-terminal β-hairpin "wing" (Fig. 1c); the ungapped alignment of this region suggests that one of these two secondary structures may have evolved into the other through stepwise mutation.

## Alignments between response regulator sequences with HTH₄ and wH folds indicate evolved fold switching

To further test whether stepwise mutations could have engendered a switch from α-helices to β-sheets (or vice versa), we next used an alternative sequence search algorithm, jackhmmer, to assess the potential evolutionary relationship between response regulators with HTH₄ and wH output domains. Although more computationally intensive, iterative Hidden Markov Model (HMM)-based searches are typically more sensitive than PSI-BLAST[47] and may better avoid homologous overextension[45]. To that end, sequences for 23 non-redundant, full-length response regulators with HTH₄ (11) and wH (12) domains were identified from the PDB using the ECOD database.

In this round of analysis, our goal was to determine whether sequences of *all* the experimentally determined full-length response regulators with HTH₄ and wH folds could be matched to sequences encoding the alternative fold (i.e., HTH₄ to wH matches, and vice versa). Using jackhmmer[47], each full-length sequence was used to query all sequences from the PDB ("Methods" section). As expected, the pairwise sequence identities of 23 full-length response regulators clustered into two subfamilies based on their CTD architectures (HTH₄ and wH, Fig. 2a), indicating that CTDs in the same fold families have closer evolutionary relationships than those in different fold families (Supplementary Fig. 1). Nonetheless, the C-terminal helices of the HTH₄ domains consistently aligned with a region in the C-terminal β-hairpin wings of wH fold domains (Fig. 2b). Furthermore, the α-helical interdomain linkers of the HTH₄ consistently aligned with the four N-terminal β-strands of the wH domain. In further support of the cross-fold relationship, another 19/34 CTD-only structures were identified by the full-length queries, again with cross-fold recognition.

The possible relationship between HTH₄ and wH folds was further supported by assessing the e-value distributions from alignments between the full-length proteins with (1) homologs from their own subfamily and (2) homologs from the alternatively folded subfamily (Fig. 2c, gray/yellow backgrounds, respectively). Median e-values of the alignments between the sequence of a given experimentally determined fold (HTH/wH) and the set of sequences with the alternative fold (wH/HTH) ranged from e-33 to e-43, suggesting significant evolutionary relationships across all members of the two subfamilies (Fig. 2c). As expected, the median e-values among sequences of similar folds ranged from e-54 to e-72 (Supplementary Fig. 2a), indicating closer evolutionary relationships.

Statistically significant alignments were also identified between full-length query sequences and isolated CTDs with the alternative fold in 22/23 full-length response regulators. Median e-values of these alignments ranged from e-04 to e-09, whereas median e-values of aligned sequences from the same fold family ranged from e-17 to e-30 (Supplementary Fig. 2b). These domain-specific alignments further support the evolutionary relationship between HTH₄ and wH domains.

Thus, the jackhmmer results (Fig. 2) are consistent with the PSI-BLAST alignment (Fig. 1b), and suggest two types of evolutionary events: (1) The linker may have been extended/shortened through an insertion/deletion; and (2) stepwise mutation may have induced a structural interconversion between the C-terminal α-helix of the HTH₄ and the C-terminal β-sheet of the wH.

## Phylogenetic analyses of HTH₄ and wH proteins

Although these structure-based sequence searches were consistent with evolved fold switching in the C-terminal HTH₄ and wH domains, the mechanism of secondary structure conversion was obscured by the alternative locations of sequences inserted into the longer wH homologs. PSI-BLAST fully aligned the C-terminal α-helix of the HTH₄ with the β-hairpin of the wH (Fig. 1b), suggesting a full secondary structure conversion. By contrast, jackhmmer aligned the C-terminal α-helix of the HTH₄ with only the first β-strand of the wH (Fig. 2b), suggesting a partial conversion along with an insertion. To discriminate between these options, we next collected a large set of response regulator sequences with HTH₄ and wH output domains. To that end, the FixJ$_{PDB}$ and KdpE$_{PDB}$ sequences were queried against the nr database using protein BLAST to identify 581,791 putative homologs. Given the size of this sequence set, we developed several strategies for curating and sampling the data ("Methods" section) so that the final subset of sequences would be small enough for various phylogenetic analyses but large enough to adequately represent the large family of response regulators.

To that end, the 581,791 sequences were grouped into 367 clusters using a greedy clustering algorithm and filtered to 85% redundancy for a final number of 23,791 sequences. Clusters were then compared to identify 13,006 FixJ-like sequences and 10,785 KdpE-like sequences. Sequences within each group readily aligned; however, the two groups had overall low sequence identities with each other. Several approaches were attempted to align these groups. One attempt identified a "transitive homology pathway" of 7 sequences connecting HTH₄ to wH sequences (Supplementary Table 3, "Methods" section) that was used to match the FixJ-like (HTH₄) and KdpE-like (wH) alignments. However, when a phylogenetic tree was constructed in IQ-Tree for the combined 23,791 sequences, its quality was poor (i.e., 140 gaps/360 positions in the KdpE$_{PDB}$ sequence) and failed to converge after 3 rounds of bootstrapping.

Nevertheless, the transitive homology path suggested the existence of additional sequences that might bridge the HTH₄ and wH fold families. Thus, we searched the original sequence set with an alternative approach. First, we categorized clusters with ≥100 sequences by their CTD architectures to identify 74,741/387,276 sequences with HTH₄/wH output domains. These sequence sets were used to construct BLAST libraries. Next, the sequences with HTH₄ output domains were filtered to 50% redundancy, and the remaining 4520 sequences

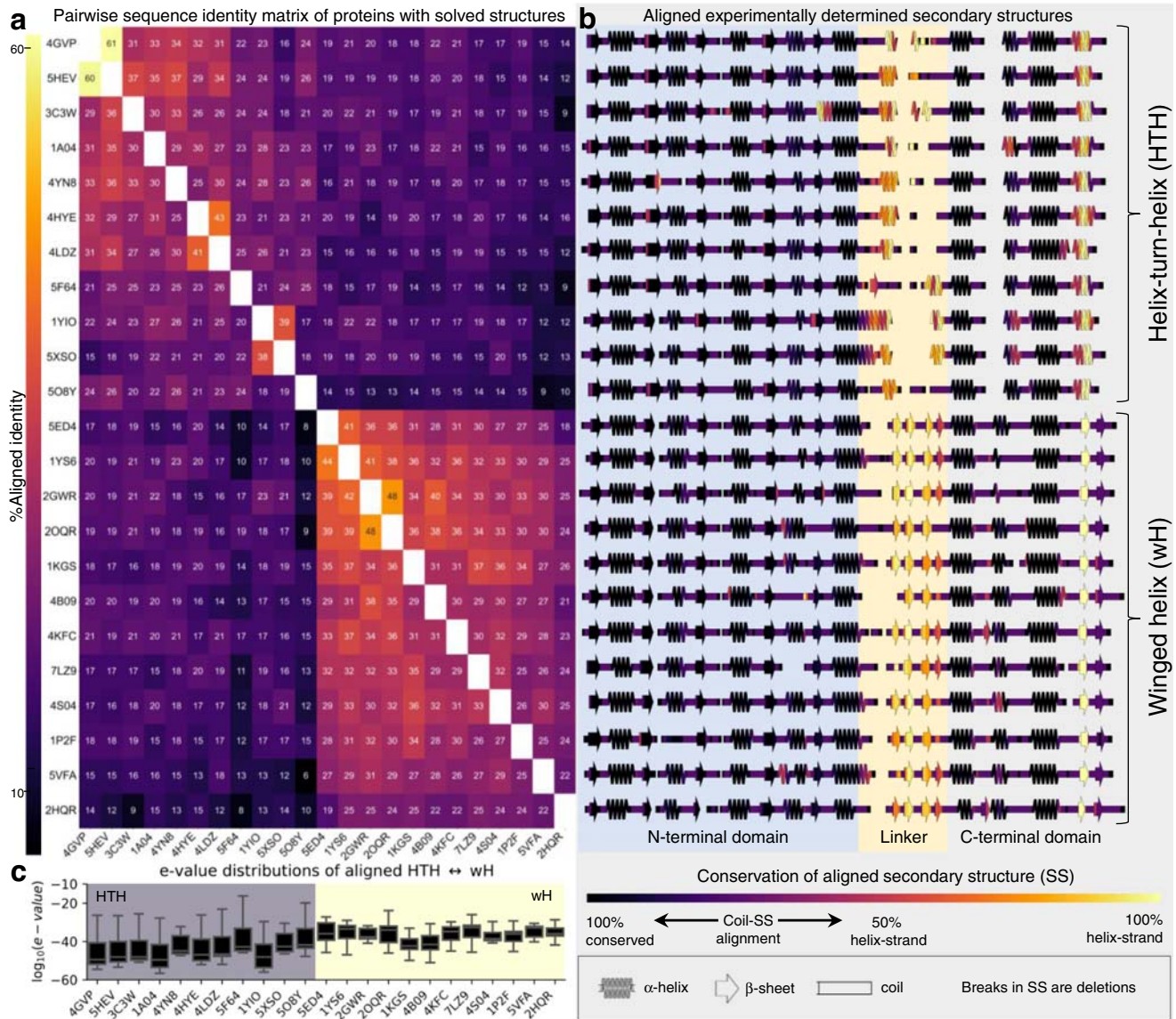

**Fig. 2 | Alignments between experimentally determined tetrahelical helix-turn-helix (HTH₄) and winged helix (wH) folds consistently indicate evolved fold switching. a** Jackhmmer-aligned sequences of response regulators with experimentally determined structures (PDB IDs) were used to calculate pairwise sequence identities. Sequences cluster into two subfamilies, with HTH₄ (upper right bracket) and wH (lower right braket) C-terminal domains. Each row reports % aligned identities (numbers within boxes) calculated from pairwise comparisons. Identical sequences are white; all others are colored by % identity (left colorbar). **b** Experimentally determined secondary structures of each sequence in **a**. The N-terminal domain, linker, and C-terminal domain are indicated by different background colors. The secondary structures are colored by their sequence-based, secondary-structure alignments with the alternatively folded structures (HTH₄ aligned with wH and vice versa). Identical secondary structures that consistently align are dark purple (e.g., helices that always align with helices); secondary

structures that align with regions of random coil range from light purple to pink; α-helices that align with β-sheets and vice versa are colored from pink to yellow, depending on whether the alignment is more or less frequent. **c** Box and whisker plots of log₁₀(e-values) of jackhmmer searches of sequences that used one fold to query sequences from the alternative subfamily (HTH₄ against wH or vice versa). The distributions of each HTH₄ (gray background)/wH (yellow background) box were derived from N = 12 (1A04, 1YIO, 3C3W, 4GVP, 4HYE, 4LDZ, 4YN8, 5F64, 5HEV, 5O8Y), 11 (1KGS, 1P2F, 2HQR, 4B09), 10 (5XSO, 2GWR, 4SO4), 9 (2OQR), 8 (4KFC, 5VFA, 7LZ9), 7 (1YS6, 5ED4) e-values; each box bounds the interquartile range (IQR) of the data (first quartile, Q1 through third quartile, Q3); medians of each distribution are gray lines within each black box; lower whisker is the lowest datum above Q1-1.5*IQR; upper whisker is the highest datum below Q3 + 1.5*IQR. Source data are provided as a Source Data file.

were queried against the wH library with protein BLAST. If a match was statistically significant, we searched NCBI sequence records of both sequences for CTD structure annotations, which are typically inferred from Hidden Markov Models. These results were used to distinguish BLAST matches between different fold families (sequence pairs with 1 annotated HTH₄ and 1 annotated wH) from matches between the same fold family. Sequence pairs with annotations from different fold families were retained; this process identified 3136 matches between 664 HTH₄ and 2541 wH proteins with mean/median e-values of

$4 \times 10^{-10}/5 \times 10^{-16}$. Reciprocal BLAST searches, using the wH sequences as queries, were successfully performed in all 3136 cases, with mean/median e-values of $1 \times 10^{-8}/2 \times 10^{-16}$; these higher e-values likely reflect the smaller size of the HTH₄ database or the longer lengths of wH sequences relative to HTH₄.

Next, we aligned the 3205 sequences using two different methods, Clustal Omega[48] and MUSCLE[49] (Supplementary Data 1). Again, a key difference between these cross-family multiple sequence alignments (MSAs) was the location of sequences inserted into/deleted from the

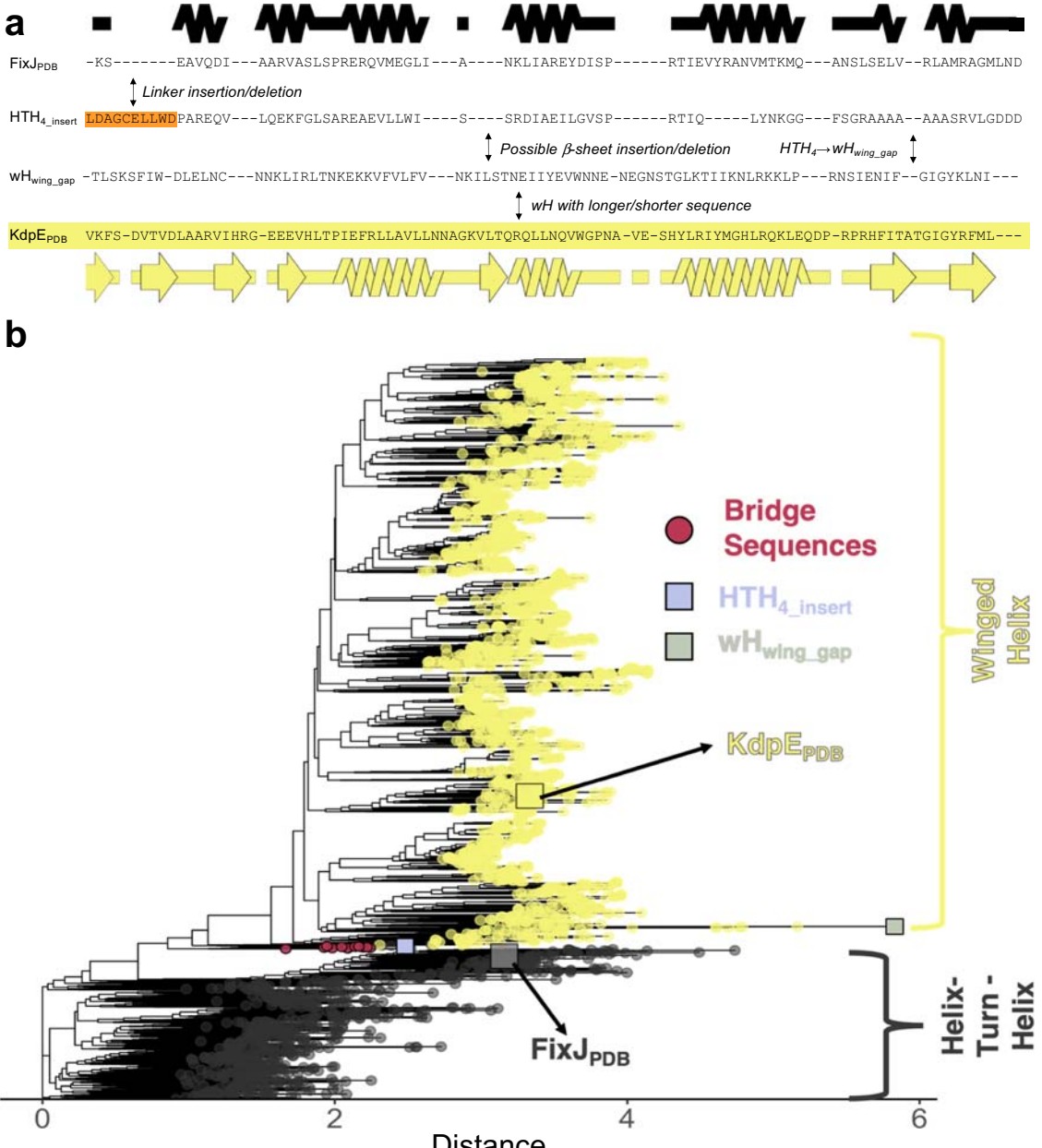

**Fig. 3 | Phylogenetic analyses identify bridge sequences adjoining families of response regulators with tetrahelical helix-turn-helix (HTH₄) and winged helix (wH) CTDs. a** Clustal Omega alignment of 3205 HTH₄ and wH sequences indicates complete conversion of C-terminal secondary structure over evolutionary history. Secondary structure diagrams were generated using the structures of FixJ$_{PDB}$ (black) and KdpE$_{PDB}$ (yellow). Background colors of the four sequences match those in the phylogenetic tree. Notes in the spaces between sequences show important changes: (1) orange linker insertion (or deletion, depending upon the properties of ancestral sequences) (2) fold conversion (3) sequence elongation/deletion. The word in front of a slash represents what happens if a sequence changes from top to bottom; the word following the slash represents what happens if a sequence changes from bottom to top. A common ancestor between the FixJ$_{PDB}$ and KdpE$_{PDB}$ sequences is also possible. Source data are provided as a Source Data file. **b** Maximum-likelihood phylogenetic trees suggest an evolutionary path between response regulators with HTH₄ and wH folds. Sequences with C-terminal domains annotated as HTH/wH from NCBI protein records are gray/yellow. The clade containing the 12 identified bridging sequences is highlighted in pink. HTH₄_insert provides an example of an annotated HTH₄ sequence whose linker length was similar to wH; wH$_{wing\_gap}$ provides an example of a wH sequence with a 2-residue deletion similar to those found in >99% of the C-terminal helices of aligned HTH₄ sequences. Distance units are arbitrary, though sequences further in space have more distant evolutionary relationships.

longer wH/shorter HTH₄ homologs. Nevertheless, in both cross-family MSAs, the C-terminal helix of the HTH₄ aligned fully with the C-terminal β-sheet wing of the wH, indicating evolution from α-helix to β-sheet by stepwise mutation rather than insertion or deletion (Fig. 3a and Supplementary Fig. 3). In the Clustal Omega alignment, a two-residue gap found in > 99% of HTH₄ folds was also found in an annotated wH fold (wH$_{wing\_gap}$), further suggesting that the α-helix ↔ β-sheet interconversion occurred through stepwise mutation.

Furthermore, several HTH₄ sequences with linker lengths similar to wH sequences were identified (e.g., HTH₄_insert in Fig. 3a), demonstrating that long linkers are not exclusive to wH folds. Sequences within the alignment were diverse, with mean pairwise identities of 31% among HTH₄ folds, 40% among wH folds, and 31% across folds. Notably, evolutionary conservation patterns differed between the HTH₄ and wH folds (Supplementary Fig. 4). Particularly, the C-terminal helix of the HTH₄ did not show strong conservation patterns, whereas the β-strand

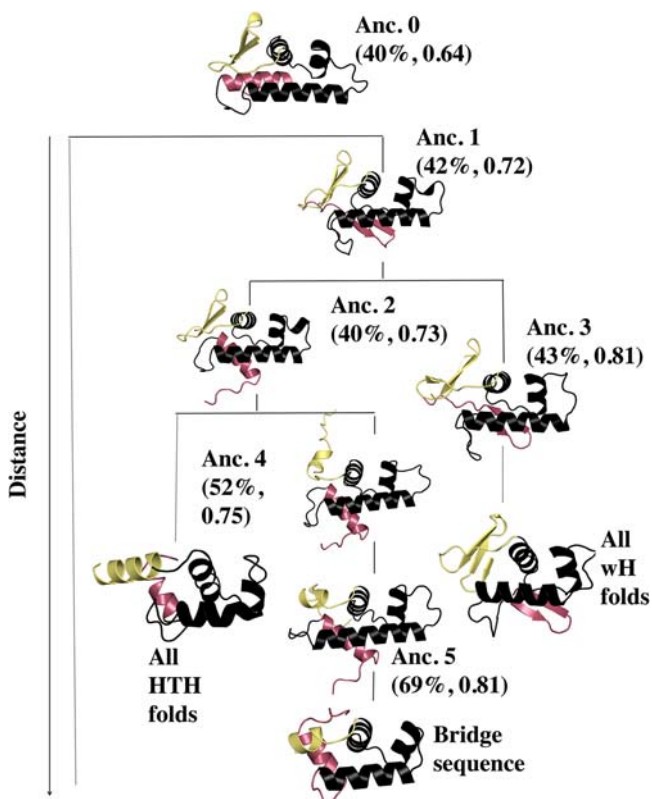

**Fig. 4 | AlphaFold2 predictions for the C-terminal domains of the reconstructed ancestors appear to switch folds in response to 2–3 mutations in the most C-terminal secondary structure element.** The earliest ancestor appears to be the longer version of a tetrahelical helix-turn-helix (HTH₄), from which winged helix (wH) folds evolved. The fold-switching C-terminal helix/β-hairpin is shown in pink, and the structurally plastic linker is shown in yellow. The bridge sequence used in this plot was TME68356.1, the one nearest the ancestral node in Fig. 3b.

wing of the wH did. As suggested by Cordes and colleagues[27], such distinct conservation patterns may explain why homology between sequences for the isolated wH and HTH₄ domains could not be inferred from the PSI-BLAST and jackhmmer searches against the PDB.

Finally, we generated a bootstrap-supported, phylogenetic tree for the cross-family MSA. Strikingly, results revealed a sequence clade that appears to bridge the two fold families (Fig. 3b and Figs. S5 and S6). The 12 sequences of this clade include one identified in the transitive homology path; all 12 have output domains annotated as HTH₄ and originated from several bacterial phyla (Supplementary Table 4). In the phylogenetic tree, these 12 sequences adjoin branches with wH and HTH₄ CTDs (Fig. 3b), suggesting that their ancestors might be evolutionary intermediates between the two folds. To assess the statistical robustness of the HTH-bridge-wH interface, we quantified the frequency of its occurrence using trees rooted in all 6393 possible branch points. The log-likelihood of each rooted tree was calculated using the approximately unbiased test[50] (p-AU, Supplementary Fig. 7A). Of the 6393 possible rootings, 18 had a p-AU score ≥0.8 (Supplementary Fig. 7B), indicating statistical significance. In all 18 cases, the bridge sequences adjoined branches with annotated wH and HTH₄ domains (Supplementary Fig. 8), strongly supporting the role of this clade as an evolutionary bridge between the two folds.

### A mutational pathway between two folds

We next examined the predicted structural properties of sequences in the bridge clade. To that end, structural models of each bridge sequence were produced with AlphaFold2[14] (AF2). Strikingly, all models assumed the HTH₄ fold (Supplementary Fig. 9). This result suggests

a few possibilities. First, some bridge sequence(s) might interconvert between HTH₄ and wH folds; previous work has shown that AF2 generally predicts only one dominant conformation of proteins that can switch between two folds[42,51]. Second, the AF2 predictions could be unreliable, and some or all bridge sequences could, in fact, assume wH folds. Thirdly, the fold transition might have occurred in earlier ancestors located at nodes linking most HTH₄ and wH sequences. These nodes connect the two fold families in the tree (Supplementary Fig. 5), suggesting that their corresponding ancestral sequences may have had properties of both HTH and wH folds.

Thus, we next performed ancestral sequence reconstruction and generated additional AF2 models for the ancestral sequences bridging the HTH₄ and wH folds (Figs. 4 and S5). Note that the linkers of all ancestral sequences were as long as the wH linkers. Our rationale was that the linkers of some HTH₄ sequences near the bridge region were equally long as the linkers of wH sequences (Fig. 3 and Supplementary Fig. 3), suggesting that these linkers may have already been modified by a large insertion.

Intriguingly, results from ancestral reconstruction suggest that the ancestor sequences may have had structurally plastic regions that could switch between α-helices and β-sheets in response to mutation (Fig. 4 and Supplementary Table 5). Notably, Ancestor 0's most C-terminal secondary structure element is an α-helix, Ancestor 1's is a β-hairpin, and Ancestor 2's switches back to an α-helix (Fig. 4, pink). Interestingly, the sequence of Ancestor 1's β-hairpin is 83% identical to the sequences of both Ancestor 0's and Ancestor 2's C-terminal helices, which are 75% identical to one another. These results suggest that just two mutations can switch the C-terminal α-helix to a β-sheet and back again through a different set of sequence substitutions.

The N-terminal linker region (Fig. 4, yellow) also appears to be plastic. In Ancestors 0–2, this linker is partially folded into a β-hairpin structure, whereas in Ancestor 3 the linker assumes a fully folded 4-β-sheet structure. In contrast, the linker assumes a partially helical structure in Ancestors 4–5 and in the modern-day bridge sequence (Fig. 4).

Taken together, these results suggest that ancestors of sequences in the bridge clade may have had propensities for *both* wH and HTH₄ folds. To further test this possibility, both PSI-BLAST and jackhmmer searches were carried out between the ancestral CTD sequences and PDB structures with both HTH₄ and wH folds. Statistically significant cross-fold matches were identified in all cases except for Anc. 3 (Supplementary Data 2). By comparison, the earlier PSI-BLAST and jackhmmer searches of the isolated CTDs of existing HTH₄ and wH sequences matched homologs with the same but not the alternative fold.

### Evolution from HTH₄ to wH may have expanded DNA-binding specificity

Finally, we sought to identify whether the shift from HTH₄ to wH folds may have had some evolutionary advantage. Examination of experimentally determined HTH₄ and wH response regulator structures in complex with their cognate DNA partners suggests that one benefit of the structural transformation might have been expanded binding specificity. On average, the HTH₄ folds contact 17 unique nucleotides, whereas the wH folds contact 22 (Fig. 5a). Both HTH₄ and wH folds have a single recognition helix that binds the major grove, and the C-terminal β-hairpin of winged helices also contacts the minor groove (Fig. 5b). As such, wH domains can likely recognize more unique nucleotide sequences than HTH.

### Discussion

Decades of research suggest that protein secondary structure is largely conserved over evolutionary history[52,53]. Accordingly, a variety of studies have shown that new protein folds can evolve through various mechanisms that keep secondary structure fixed, such as insertions, deletions, and circular permutation[54]. Others have shown that proteins

**a**

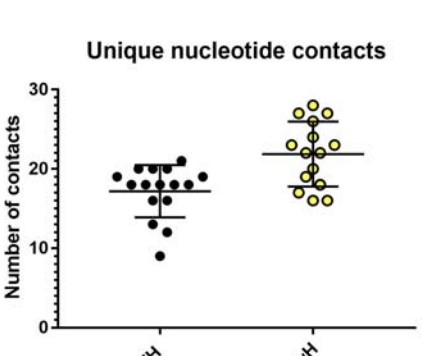

**b**

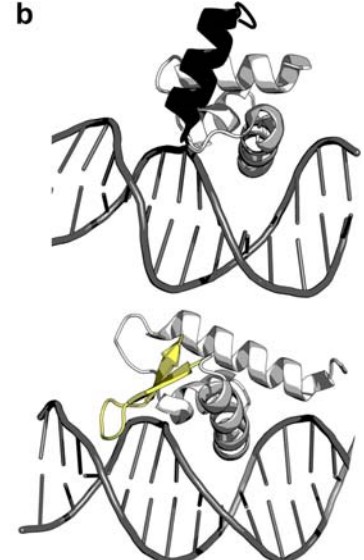

**Fig. 5 | Tetrahelical helix-turn-helix (HTH₄) domains contact fewer nucleotides, on average, than winged helix (wH) domains. a** Simplified box-and-whisker plot with overlaying datapoints for the number of contacts between HTH₄ and DNA (black) and wH and DNA (yellow). On average, HTH₄ domains have 5 fewer DNA contacts than wH domains. Central bars correspond to means, upper/lower bars to standard deviations. Statistics were derived from 16/15 independently determined structures of HTH-DNA/wH-DNA complexes. Source data are provided as a Source Data file. **b** Examples of DNA (gray) interactions with HTH₄ and wH domains, above and below, respectively. The C-terminal α-helix of the HTH₄ (black, above) does not contact the DNA, whereas the β-hairpin wing of the wH (yellow, below) contacts the minor groove. Structurally similar parts of the HTH₄ (PDB ID: 1h0m, chain D) and wH (PDB ID: 4hf1, chain A) folds are light gray. This result and the corresponding increase in the possible number of unique DNA sequences that could be recognized by the wH might explain why it evolved from the HTH₄ in response regulators.

with conserved secondary structures can evolve different tertiary arrangements[24,25,55].

In contrast, several recent studies suggest that stepwise mutations can switch protein secondary structures, fostering the evolution of new protein folds[19,28,56,57]. Our work supports this hypothesis by identifying a statistically significant evolutionary trajectory between two protein folds. These folds comprise fragments of response regulator CTDs that switch from α-helix to β-sheet. Our findings are supported by ancestral sequence reconstruction, structural models, and several sequence alignment methods. Furthermore, this evolved fold switching likely had a functional consequence: expanding DNA-binding specificity. Notably, HTH₄ and wH folds are not limited to the superfamily of response regulators. In other families, the wHs could have evolved from HTH₄ ancestors through different or additional mechanisms (and the evolutionary order may differ).

Since the fold-switching region observed here comprises a fragment of the whole protein, we compare our proposed stepwise mechanism to other mechanisms for protein evolution that involve protein fragments, such as "words"[58] and "bridging themes"[59–61]. The work presented here differs from these studies in several important ways. First, "words" were defined as protein fragments with "local similarities in sequence and structure within globally different folds"[58], and bridging "themes" each comprise a set of "homologous protein fragments found in different sequential and structural contexts"[59]. As such, the isolated sequences of these fragments have discernable homology without the context of the rest of the protein. In contrast, the fold-switching sequences of HTH₄ and wH fragments reported here only exhibited discernible homology within the context of the whole protein. Practically speaking, the searches used to identify words and themes, which rely on matches between homologous sequence of protein fragments in different protein contexts, could not be used to identify the evolved fold switching transition proposed here.

Second, the evolutionary mechanism underlying words and bridging themes differs from the stepwise mutation that likely caused the HTH₄ domains of response regulators to evolve into wH folds. Words and bridging themes are conserved protein fragments proposed to either recombine with or accrete non-homologous segments of protein structure to form distinct domains. In contrast, the fold-switching transition proposed here occurs within a conserved protein context. In this case, stepwise mutations appear to have caused a protein fragment to switch from α-helix to β-sheet without fragment recombination or accretion. Importantly, fragment recombination, accretion and stepwise mutation are all valid evolutionary mechanisms that occur in different situations.

Third, although some bridging themes switch folds[61], their switching likely depends on their larger protein context. That is, within differently folded domains, the same bridging theme may also assume different folds. This is also true of chameleon sequences[62,63], identical protein fragments with different folds in different protein contexts. In contrast, the homologous sequences in this work assume different structures within homologous protein contexts: both folds are C-terminal to a conserved trihelical helix-turn-helix[39]. It cannot be overstated that the fold switch we report was covert: homology between the sequences of the fold-switching region could not be identified without the context of the rest of the protein, including the N-terminal receiver domain. This critical point distinguishes our findings from previous studies of words and bridging themes, as well as from the "creative destruction" mechanism by which new folds evolve through fusions of genes encoding distinct domains[64].

Although outside the scope of this study, experimental testing of the reported bridge sequences and reconstructed ancestors may reveal mechanistic details of the transition from HTH₄ to wH. Whether any of these sequences populate both folds – as has been observed for other fold-switching proteins[57,65]– would be of particular interest. For the reconstructed ancestors, structural interconversion would be analogous to functional studies of reconstructed ancestors of green and red fluorescent proteins that emit both green and red light[66] or promiscuous glucocorticoid receptors reconstructed from extant receptors with unique binding specificities[67]. As previous work has shown[57,68,69], structural interconversion can be observed with nuclear magnetic resonance (NMR) spectroscopy. Indeed, NMR studies of the Arc repressor[70,71] and XCL1[57] identified a handful of key mutations that

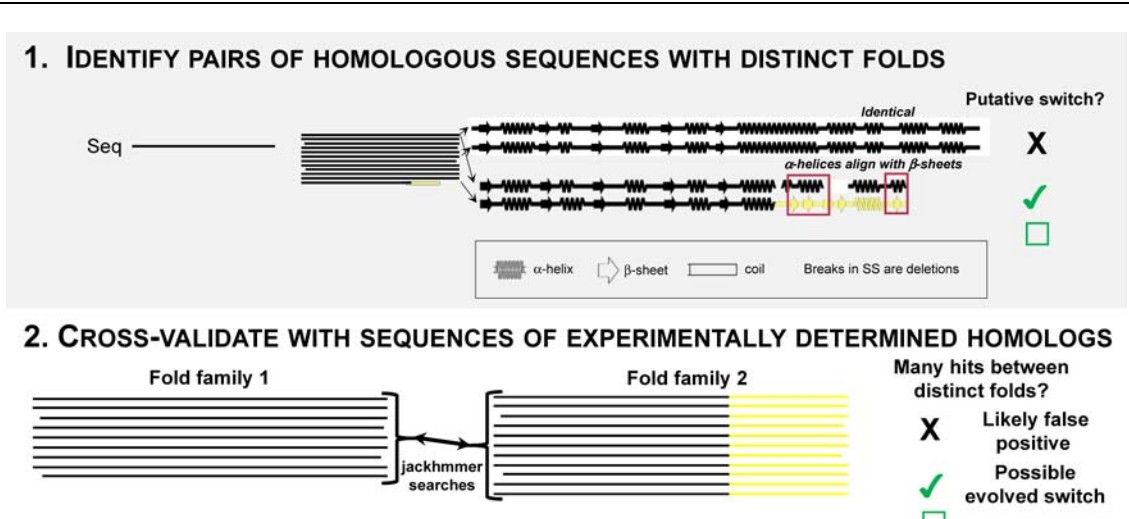

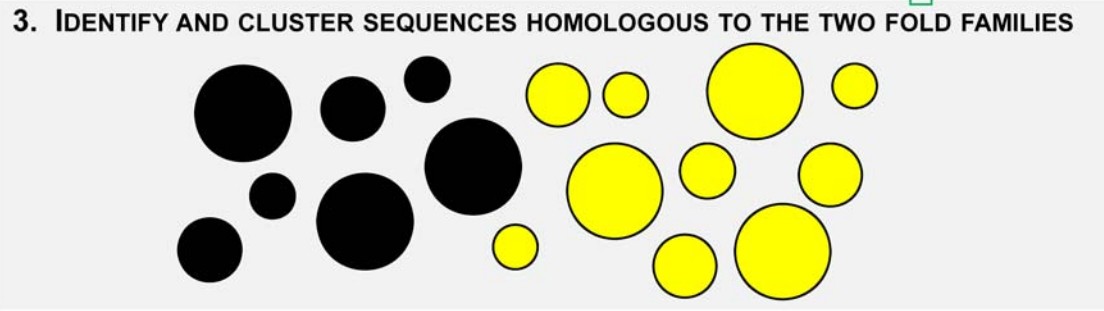

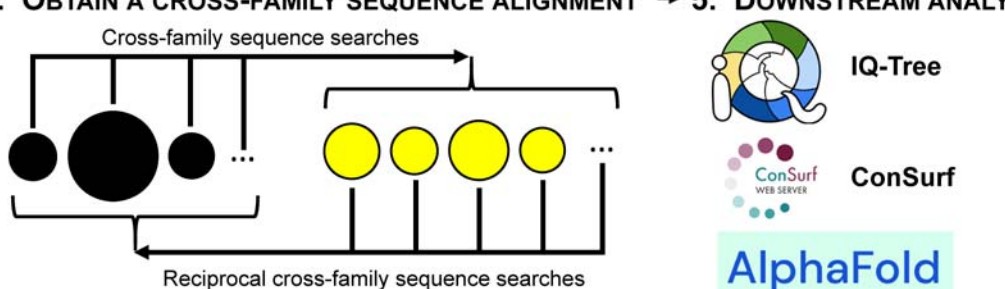

**Fig. 6 | A step-by-step guide to identifying more evolved fold switchers.** 1. Query a sequence of interest (black) against the PDB (or database of predicted structures) with one round of protein BLAST (or phmmer) and search for hits with distinct secondary structures (yellow). Hits may indicate evolved fold switching. 2. Cross-validate results from step 1 by performing more sensitive sequence searches (e.g., jackhmmer) of all homologous sequences with experimentally determined structures. Black sequences=Fold1; yellow sequences=Fold2. Black regions of Fold2 have the same folds as Fold1 to allow for the possibility that Fold2 is a protein sub-domain. 3. If cross validation is successful, find all sequences homologous to Fold1 (black) and Fold2 (yellow); cluster sequences by likely fold family. 4. Obtain a cross-family sequence alignment by searching all sequences from Fold1 against Fold2 and reciprocally searching Fold2 hits against Fold1. 5. Use cross-family alignment for downstream analyses including, but not limited to, IQ-Tree, ConSurf, and Alpha-Fold2. Complete descriptions of each step can be found in the main text.

switch protein folds. Accordingly, it would be interesting to experimentally identify minimal mutational pathways that switch HTH$_4$ sequences to wH and vice versa.

Biophysically based computational approaches may also provide insights into the mechanism and evolution of response regulator proteins with HTH$_4$ and wH domains. Such studies successfully predicted fold transitions between engineered protein G variants with high levels of sequence identity but different folds[72,73]. Other biophysical models or hybrid theoretical-experimental approaches can be used to infer the roles of point mutations, multifunctionality, selective pressure, and epistasis in protein evolution[74–76].

Secondary structure switching, such as the instance identified here, may be more common in the evolutionary record than currently realized. Among our results, an evolutionary pathway from HTH$_4$ to wH was consistently observed, with a clade of "bridge sequences" occupying a key location in the pathway. Notably, these bridge sequences were identified from metagenomic sequencing performed primarily in 2018 and 2019, which demonstrates the importance of new sequencing techniques and initiatives for advancing evolutionary studies[77] and suggests that more instances of evolved fold switching might now be identifiable.

Thus, we close by offering the following step-by-step guide (Fig. 6) to aid future computational searches for evolved fold switching:

(1) Identify pairs of homologous sequences with distinct folds. Here, we achieved this by performing an all-against-all search of the PDB using protein BLAST (Fig. 1, "Methods" section). Additional instances of evolved fold switching may be identified as more structures are deposited. Alternatively, structural models generated by predictive algorithms such as AlphaFold2[14], ColabFold[78], RGN2[13], or ESM-fold[79] could be used instead of experimentally determined predictions. Though less certain than experiment, these predicted structures could provide useful starting points for

sequence analyses and subsequent experimental testing. Notably, successful identification of the evolutionary pathway reported here required that the whole protein sequence be searched (N-terminal+C-terminal domains) rather than the fold-switching C-terminal domain only.

(2) Cross-validate findings using homologous sequences with experimentally determined structures. Here, we performed jackhmmer searches of all response regulator sequences with $HTH_4$ and wH domains whose structures had been experimentally determined. We found signs of cross-fold homology for all sequences (Fig. 2). This approach provides confidence that the evolutionary relationship identified in Step 1 spanned both protein families rather than being a single hit obtained by chance. Similar analyses could be performed on experimentally determined structures of putative evolved fold switchers from other protein families. If such structures are not available, they could be generated using predictive algorithms. If many predictions need to be made, we recommend using ColabFold[78] because of its high accuracy and superior performance.

(3) Identify and cluster sequences homologous to the two fold families. *Identify.* For the FixJ/KdpE sequences, BLAST searches of the nr database returned >1,000,000 sequences. We used BLAST because of its efficiency in searching such a large database, though a more sensitive high-efficiency method such as HHBlits[80] could also be used. Curation of the sequence set ("Methods" section) may be required to remove anomalous sequences. *Cluster.* Although we used a custom-written greedy clustering algorithm, MMSeqs2[81] could also be used. Next, we associated each remaining cluster with a given fold by BLASTing the sequences of $FixJ_{PDB}$ and $KdpE_{PDB}$ against each cluster and calculating which sequence yielded more matches with ≥200 residues and *e*-values ≥ 1e-04.

(4) Obtain a cross-family sequence alignment containing sequences with high e-values but different structural annotations. For successful completion of downstream analyses, this "Goldilocks" step is key: overly large alignments can lead to uninterpretable results ("Methods" section), but alignments that are too small could inadvertently omit important evolutionary intermediates. For this work, we extracted and constructed the relevant cross-family alignment by searching all sequences from clusters assigned to one fold (Fold1) against all sequences from clusters assigned to the other (Fold2). Since this process involved thousands of independent searches, protein BLAST was used for efficiency. For consistency, reciprocal searches of Fold2 matches against the Fold1 database are necessary. As a final validation step, it is advisable to discard sequences from Fold1/Fold2 clusters that were not annotated as Fold1/Fold2 in their NCBI sequence records. The remaining set of cross-family sequences can then be aligned using multiple algorithms. In this case, we used both Clustal Omega[48] and MUSCLE[49].

(5) Perform downstream phylogenetic analyses. Here, we did phylogenetic analyses on our cross-family sequence alignment with IQ-Tree[82] and Consurf[83] and ancestral sequence reconstruction with IQ-Tree. A cross-family alignment should be compatible with a range of other phylogenetic analysis methods.

## Methods
### BLAST and PSI-BLAST searches of the PDB
To identify the putative evolutionary relationship between $FixJ_{PDB}$ and $KdpE_{PDB}$, we performed protein BLAST searches with maximum e-value of 1e-04 on all sequences within the Protein Data Bank (PDB) against all other PDB sequences[16,41]. To determine whether homologous sequences folded into different structures, secondary structure annotations of each PDB, by DSSP[84], were aligned in register with their corresponding BLAST alignments and compared one-by-one, position-by-position. This approach allowed us to quantitatively assess the

similarity of aligned secondary structures. A potential match was required to have a continuous region of at least 15 residues in which at least 50% of the residues showed α-helix ↔ β-sheet differences. Using this approach, the sequence of $FixJ_{PDB}$ matched the sequence of $KdpE_{PDB}$ with an e-value of 1e-07; differing secondary structures in the C-terminal output domains were identified through DSSP comparison. Subsequent three-round PSI-BLAST searches of $FixJ_{PDB}$ and $KdpE_{PDB}$ sequences against all PDB sequences were performed with a gap open penalty of 10 and a gap extension penalty of 1. In CTD PSI-BLAST searches, the sequences for $FixJ_{PDB}$ and $KdpE_{PDB}$ spanned residues 124–205 and residues 129–225, respectively. Importantly, $FixJ_{PDB}$ and $KdpE_{PDB}$ were defined to have different folds by several independent annotators: Pfam[85] (http://pfam.xfam.org): PF00010 (helix-turn-helix), PF02319 (winged helix). ECOD[86] (http://prodata.swmed.edu/ecod/) puts them in different T-groups (tetrahelical HTH and winged), SCOP[10] (https://scop.mrc-lmb.cam.ac.uk): HTH: 8034563 (Superfamily C-terminal effector domain of the bipartite response regulators) Winged helix: 8075578 (Superfamily: PhoB-like).

### jackhmmer alignments of structures with response regulator sequences
To test the PSI-BLAST results obtained previously, jackhmmer searches were also performed on $HTH_4$ and wH sequences with experimentally determined structures. Accordingly, structures of 23 full-length response regulators with $HTH_4$ (11) and wH (12) output domains were identified from the Evolutionary Classification of Protein Domains (ECOD) database[86]. Five rounds of jackhmmer were run on each of the 23 sequences with gap open/extension probabilities of 0.05 and 0.5, respectively, using a database of all sequences downloaded from the PDB (7/15/2021) and removing sequence duplicates post-search. Sequence identities from each row of Fig. 2a were calculated from each sequence alignment generated by jackhmmer run on the sequence of the PDB entry with ID labeling each respective row.

DSSP annotations were aligned in register with each jackhmmer-generated sequence alignment to compose the secondary structure diagrams in Fig. 2b. In further detail, secondary structure annotations of each of the 11 $HTH_4$s were compared with secondary structure annotations of 48 wHs identified from ECOD; likewise, secondary structure annotations of each of the 12 wHs were compared with secondary structure annotations of 35 $HTH_4$s identified from ECOD (Supplementary Data 3). Similarities of each pair of aligned secondary structure (46 pairs for each of the 11 $HTH_4$ proteins, 30 pairs for each of the 12 wH proteins) were scored as follows: +1 for a position with identical secondary structures (helix:helix [H,G,I in DSSP notation] or strand:strand [E in DSSP notation]) and −1 for a position with alternative secondary structures (helix:strand or strand:helix using the same DSSP notations as above). Position-specific scores were normalized by the frequency of ungapped residue pairs in each position, including coil-secondary structure alignments, effectively scored as 0. These normalized position-specific scores were used to generate the colormaps of each secondary structure diagram.

### Identifying large sets of response regulators' genomic sequences
The full sequences of both $FixJ_{PDB}$ (PDB ID 5XSO, chain A) and $KdpE_{PDB}$ (PDB ID 4KFC, chain A) were searched against the nr database (10/8/2020) using protein BLAST with a maximum e-value of 1e−04 and a maximum of 500,000 alignments per search. Full sequences from each alignment were retrieved by their NCBI accession codes using blastdbcmd on the nr database. All sequences from both searches were combined, which totaled 999,912 after sequence duplicates were removed. Sequences with either fewer than 162 or more than 300 residues were removed because they likely lacked the proper response regulator domain structure, leaving 581,791 sequences. This was too many to curate using standard tools, and many sequence identities

were well below the ~40% identity threshold, below which many alignment tools become unreliable[87]. Thus, to further analyze these sequences, we performed the clustering and sampling methods described in the following sections.

### Generating sequence clusters

From set of 581,791 sequences, a basis set of 367 sequences – each with <24% pairwise identity to all other members of the set – was selected to seed sequence clustering. Above this threshold, response regulator sequences would be expected to assume similar structures[52]. To identify this set of seed sequences, the first sequence in the list of 581,791 sequences (FixJ$_{PDB}$) was chosen. Subsequent sequences were aligned with FixJ$_{PDB}$'s sequence using Biopython[88] pairwise2.align.localxs with gap open/extension penalties of −1, −0.5, respectively. If a sequence's pairwise identity with the FixJ$_{PDB}$ sequence <24%, it was added to the basis set. Sequences in the list were aligned with all sequences previously added to the basis set and included only if the identities of all pairwise alignments were <24%, yielding 367 total basis sequences. The remaining 581,424 sequences were clustered with the basis sequence to which they had the highest aligned pairwise identity, determined exhaustively by aligning all sequences with all basis sequences using pairwise2.align.localxs, with parameters as before.

To further reduce the total number of sequences, we disregarded the 251 clusters with fewer than 50 sequences. The remaining 116 clusters comprised 103 "medium" clusters (<5000 sequences) and 13 "large" clusters (> 4000 sequences). Of the large clusters, one contained the sequence of FixJ (PDB ID 5XSO) and 283,762 other sequences, and another contained the sequence of KdpE (PDB ID 4KFC) and 25,035 other sequences.

### Curating sequence clusters

**Medium clusters.** Sequences within each medium cluster were first aligned using Clustal Omega[48]. Visual inspection revealed that some alignments were biased by sequences that were either substantially shorter or longer than majority of the homologs in their cluster. To computationally identify and filter out such sequences, we identified (i) "sparse zones" by searching for windows of 8 positions where more than 95% of the sequences contained gaps, and (ii) "populated zones" by searching windows of 10 positions where more than 90% of the sequences contained amino acid residues. Sequences with (1) ≥10% of their amino acids in sparse zones or (2) <10% of their amino acids in populated zones were removed from the cluster. The 10% thresholds were determined empirically to best perform this "culling" step. Next, we performed ~2–7 successive iterations of culling and Clustal Omega alignments, until the number of sequences in each cluster converged. During this process, 9 medium clusters shrunk to fewer than 50 sequences and were subsequently ignored, leaving 94 medium clusters.

Finally, since Clustal Omega's global alignment algorithm does not accurately report phylogeny or suggest structure, the multiple sequence alignments were further aligned using PROMALS[89], which first groups sequences based on phylogeny and then performs local alignment of recognized structural domains. The quality of all cluster alignments was inspected visually.

**Large clusters.** The large clusters, with thousands of sequences, required different strategies to appropriately generate a subsample that was tractable for additional sequence analyses. To determine subsample sizes that adequately represented the sequence composition within clusters, three independent, random subsamples of 1000 and 5000 sequences were extracted from the FixJ cluster, and three 5000 sequence subsamples were extracted from the KdpE cluster. These subsamples were subjected to iterative culling and alignments like the medium clusters (described above).

Next, the multiple sequence alignments (MSAs) of these subsamples were uploaded to ConSurf[83] (https://consurf.tau.ac.il/consurf_index.php). Resulting scores were compared to determine how many sequences were required to give consistent evolutionary rates. Results indicated that 5000 sequences were required for an adequate representation of the both the FixJ and KdpE clusters. Visual inspection of heatmaps generated from sequence identity matrices of these sequence alignments supported the conclusion that 5000 sequences evenly sampled the sequence space. Thus, to represent the FixJ and KdpE clusters, we randomly chose one of its 5000 subsamples sequence sets. For 8 of the 11 large clusters with >5000 sequences, we similarly subsampled 5000 sequences. The 3 large clusters with <5000 sequences were curated as described for the medium clusters.

### Constructing FixJ and KdpE-specific MSAs

The high sequence diversity between clusters, with cross-cluster pairwise aligned sequence identities often <24%, impeded MSA assembly of the FixJ-KdpE superfamily. Thus, we looked for strategies to assemble sequences from the 94 medium clusters, 11 large cluster subsamples, and the 5000-sequence subsamples of the FixJ and KdpE large clusters into one combined MSA. First, we classified the clusters into two half-families with sequences resembling those in either the FixJ or KdpE large clusters. To that end, we matched sequences from each cluster with all sequences from the FixJ and KdpE large clusters with protein BLAST. Sequences from these clusters tended to align with high statistical significance to one of the large clusters but not both, simplifying cluster classification. This approach showed promise because sequences from each cluster aligned to sequences from other clusters with identities ≥38%, fostering reliable alignments. After completing all BLAST searches, 45 medium and 6 large clusters were assigned to the FixJ half-familiy for a total of 13,006 sequences and 49 medium and 5 large clusters to the KdpE half-family for a total of 10,785 sequences.

Despite sampling and curation, both half-families were too large to create an MSA using conventional tools. Thus, we used an alternative approach in which two reference alignments were generated using Clustal Omega to align representative sequences from each cluster (51 sequences for FixJ and 54 for KdpE). PROMALS was then used to refine the two half-family reference MSAs. Upon visual inspection, 7 sequences were removed from the KdpE reference MSA because they generated many gaps in the alignment; their clusters of origin were subsequently ignored. The remaining sequences in the KdpE reference MSA were realigned using Clustal Omega and PRO-MALS. Finally, upon visual inspection, the registers of prolines and charged amino acids were manually edited to match in 3 sequences (PSQ94266, HBD38673, and KEZ75144) between the registers 225 and 270 in the KdpE reference MSA. No such manual curation was needed in the FixJ MSA. Sequences within each of the remaining 98 clusters were then (i) independently aligned with PROMALS and (ii) integrated into the appropriate half-family reference MSA using MARS (Maintainer of Alignments using Reference Sequences for Proteins[90]). The MARS program allows curated sequence alignments with at least one sequence in common to be merged with each other without re-aligning the whole sequence set. Using this program, all sequences of the 51 FixJ-matching clusters and the curated subsample of the FixJ cluster were merged, using the FixJ half-family reference MSA as a guide. Similarly, all sequences of the 47 KdpE-matching clusters along with the curated subsample of the KdpE cluster were merged.

### Constructing a FixJ-KdpE superfamily MSA

The pairwise identities of sequences across the two half-families were too low to reliably create an MSA. Thus, we tried a "transitive homology" approach to combine the half-family alignments into one alignment for the superfamily. First, we identified a "path" of related sequences[91,92] following the logic that, if sequences A and B are

homologous and sequences B and C are homologous, then homology between sequences A and C can be assumed through the "bridge" sequence B. To carry out this strategy, we used protein BLAST to search for the highest sequence identity match between the unsampled FixJ and the KdpE large clusters (i.e., the clusters with >250,000 and >25,000 sequences). This hit was then queried against the database of the opposite fold and so on until we identified 7 sequences with pairwise sequence alignments each with ≥38% sequence identity that connected the FixJ sequence to the KdpE sequence (Supplementary Table 3). Note that the "bridge" sequence TME68356 (Supplementary Table 4) could align well with another sequence in either half-family, although it was originally assigned to the KdpE half-family. The top/bottom four sequences in Supplementary Table 3 were aligned with the FixJ/KdpE half-families using Clustal Omega. We next used MARS to combine half-family alignments using the bridge sequence as the reference. The resulting whole family MSA contained 45,199 sequences. These sequences were filtered to 85% redundancy with CD-HIT, ultimately yielding an MSA with 23,791 sequences. However, when a phylogenetic tree was constructed in IQ-Tree for this sequence set, its quality was poor (i.e., 140 gaps/360 positions in the $KdpE_{PDB}$ sequence) and failed to converge after 3 rounds of 1000 bootstrapping iterations each.

### Constructing a cross-family MSA

The transitive homology path identified above (Supplementary Table 3) suggested the existence of additional sequences that might bridge the $HTH_4$ and wH folds. Accordingly, the five/six previously-assigned FixJ/KdpE sequence clusters with >4000 sequences were each combined and converted into two BLAST databases representing $HTH_4$ (FixJ-like) and $wH_4$ (KdpE-like) sequences. Sequences within the combined FixJ sequence clusters were reduced to 50% redundancy using CD-HIT[93] with a word size of 2, as recommended. Protein BLAST searches were performed on each of the remaining 4520 sequences with a maximum e-value of 1e−04 using the full $KdpE_{PDB}$ database. All 8607 alignments with minimum sequence identities and lengths of 33% and 200 residues, respectively, were considered significant. To ensure that these alignments truly matched $HTH_4$ with wH sequences, NCBI records of 1793 $HTH_4$, and 4995 wH sequences were retrieved using NCBI's efetch. Each record was searched for structural annotations of its CTD (HTH or wH). Ultimately, 3074 BLAST matches, each with one annotated HTH and one annotated wH CTD were retained.

To identify additional HTH sequences that might match with wH sequences, additional BLAST searches were run on all 4 $HTH_4$ sequences in our set of 3074 matches that aligned with wH sequences with ≥38% pairwise identity. This time, the database comprised all 581,791 length-limited sequences identified from the initial FixJ and KdpE BLAST searches. These searches, intended to identify additional $HTH_4$ sequences regardless of how they were clustered, yielded 66 putative HTH sequences that might match well with additional wH sequences. Finally, 66 additional Protein BLAST searches were performed by querying each of the 66 putative HTH sequences against all sequences from the 47 KdpE-matching clusters identified previously. The resulting 62 matches with minimum sequence identities and lengths of 33% and 200 residues and HTH/wH annotations from their NCBI records, identified as before, were included, totaling 3136 matches between 3203 sequences. For reference, the sequences of $FixJ_{PDB}$ and $KdpE_{PDB}$ were also included; these two sequences had minimum aligned identities and lengths of 32% and 198, respectively, to sequences encoding the alternative folds.

The resulting 3205 sequences were aligned in two ways, with Clustal Omega and with MUSCLE[49] version 3 using the super5 command. Columns with >75% gaps were removed from both alignments using Geneious Prime 2022.2.2 (https://www.geneious.com) for further analyses. The final alignments showed full overlap between the C-terminal helix of the $HTH_4$ and the β-hairpin wing of the wH.

Subsequent phylogenetic analyses and ancestral sequence reconstruction were performed on the Clustal Omega alignment.

### Conservation scores and rate of evolution

A version of ConSurf that could be run locally, Rate4Site 2.01[94] (https://www.tau.ac.il/~itaymay/cp/rate4site.html), was used to compute evolutionary rates for the full alignment of 3205 sequences as well as the separate $HTH_4$ and wH subfamilies (664 and 2541 sequences, respectively; Supplementary Fig. 4). This program requires an MSA file to compute a phylogenetic tree. We chose the empirical Bayesian method to generate the rates, which significantly improves the accuracy of conservation scores estimations over the Maximum Likelihood method[94]. The scores are represented as grades ranging from conserved (9) to variable (1).

### Phylogenetic analyses of the cross-family MSA

**Constructing a maximum-likelihood tree and performing bootstrapping.** A maximum-likelihood (ML) phylogenetic tree was inferred from the alignment with FastTree[95,96], using the Jones-Taylor-Thorton/JTT[97] models of amino acid evolution and the CAT[98] approximation to account for the varying rates of evolution across sites. This tree was further supported by ultrafast bootstrapping (UFBoot)[99] as implemented in IQ-Tree2[82]. We used ModelFinder[100] to identify the best fitted evolutionary model for the MSA (chosen model - LG + F + R10), and then evaluated branch support with 1000 UFBoot replicates. The minimum correlation coefficient for the convergence criterion was set at 0.99. A consensus tree was also generated (Supplementary Fig. 5).

**Rooting the phylogenetic tree.** The ML and consensus trees generated by FastTree and IQ-Tree2, respectively, lacked information on root placement of the estimated phylogeny. Ideally, external information – such as an outgroup – is used to root the tree. However, we could not use an outgroup because it was not possible to identify a single sequence outside of our alignment that was homologous to both folds. Therefore, we combined the nonreversible model with a maximum-likelihood model[101] used to calculated the log-likelihoods of the trees being rooted on every branch of the tree. Bootstrapping of 10,000 replicates was performed to obtain reliable results. The method returns a list of 6393 trees rooted at each node and sorted by log-likelihoods in descending order, along with other scores by different tests, as follows; bp-RELL: bootstrap proportion using RELL method[102], p-KH: p-value of one-sided Kishino-Hasegawa test[103], p-SH: p-value of Shimodaira-Hasegawa test[104], c-ELW: Expected Likelihood Weight[105] and the p-AU: p-value of approximately unbiased (AU) test[50].

The AU test uses a newly devised multiscale bootstrapping technique developed to reduce test bias and to obtain a reliable set of statistically significant trees. The AU test, like the SH test, adjusts the selection bias overlooked in the standard use of the bootstrap probability and KH tests. It also eliminates bias that can arise from the SH test[50]. Overall, the AU test has been shown to be less biased than other methods in typical cases of tree selection and is recommended for general selection problems[50]. Hence, we relied on p-AU (p-values from AU) to get a list of 18 most-likely rooted trees with p-AU > 0.8.

**Ancestral sequence reconstruction.** Ancestral sequence reconstruction was performed using maximum-likelihood methods implemented in IQ-Tree2, which uses the algorithm described in Yang et al.[106]. Ancestral sequences were determined for all nodes of the consensus tree (Supplementary Fig. 5) using the empirical Bayesian method. Posterior probabilities are reported for each state (amino acid) at each node. We scored the nodes in three steps. First, we calculated the average probability considering all assigned states at the node. Then, replacing the states by the amino acids in the bridge sequence (TME68356.1), we calculated the total p-value. Finally, calculated the pairwise sequence identity between ancestral sequence

and the bridge sequence. Using all three criteria, we identified 6 reconstructed sequences with low *p*-values near the bridge sequences. These sequences were used for downstream analysis and model building.

## Predicting structures of ancestral and bridge sequences

The FASTA sequences of the 6 reconstructed ancestors, along with the 12 bridge sequences, were used as input to the full build of the AlphaFold2.1[14] structure prediction model. MSAs were generated by the default procedure of combining sequence searches of the BFD, MGnify, and Uniref databases. Predictions were made using templates with a maximum date of 4/20/2022. Structures ranked 0 were depicted in Fig. 4 and S9. To test the plausibility of the AF2-generated structures for the reconstructed ancestors and bridge sequences, we examined recently released AF2 predictions for 338 HTH$_4$ and 937 wH sequences[107]. AF2 predictions matched genomic annotations in every case. Prediction qualities varied: of 1275 predicted structures, 29% were predicted with high confidence, 58% had moderate confidence, and the remaining 13% had low confidence.

## Counting protein-DNA contacts

The unique nucleotide contacts between the response regulators and their corresponding DNA sequences were identified using Resmap[108], a tool that uses the atomic coordinates from PDB files to calculate intra-atomic distances for non-covalent interactions under set thresholds. The default distance thresholds for different interaction types that were used are: (1) Hydrogen bonds - ≤3.5 Å, (2) Hydrophobic interactions - ≤4.5 Å, (3) Aromatic interactions - ≤4.5 Å, (4) Destabilizing contacts - ≤3.5 Å, (5) Ion pairs - ≤5.0 Å, (6) Other contacts (which include van der Waals interactions) - ≤3.5 Å. Since the nomenclature for DNA atoms has changed since the development of Resmap, the PDB files were manually edited to match Resmap's input format with the following changes: (1) Symbol replacements of ' to *, (2) the nucleotide atoms (A,C,G, or T) were appended with the prefix 'D' (DA, DC, DG, DT), (3) the edited nucleotide atoms were also assigned unique atom identification numbers. The PDB files with these changes were then inputted into Resmap to identify unique contacts between of atoms in the protein chains with the atoms in DNA chains.

## Scripts and figures

Protein figures were generated in PyMOL (The PyMOL Molecular Graphics System, Version 2.0 Schrödinger, LLC) (https://pymol.org/2/), plots and heatmap in Matplotlib[109] (https://matplotlib.org/stable/index.html) and seaborn[110] (https://seaborn.pydata.org/). Phylogenetic trees were visualized with ggtree (https://guangchuangyu.github.io/ggtree-book/chapter-ggtree.html) implemented as an R package[111].

## Reporting summary

Further information on research design is available in the Nature Portfolio Reporting Summary linked to this article.

## Data availability

The data generated in this study, including sequence alignments and clusters, phylogenetic analyses, and AlphaFold2 models, have been deposited in the Zenodo database under accession code https://doi.org/10.5281/zenodo.7837636. The supporting data generated in this study are provided in the Supplementary Information and the Source Data file. The structural data used in this study are available in the Protein Data Bank (PDB) under accession code 5XSO, [https://doi.org/10.2210/pdb5SXO/pdb], chain A (FixJ$_{PDB}$) 4KFC, [https://doi.org/10.2210/pdb4KFC/pdb], chain A (KdpE$_{PDB}$), 1H0M [https://doi.org/10.2210/pdb1H0M/pdb], chain D, and 4HF1 [https://doi.org/10.2210/pdb4HF1/pdb], chain A. The structure classifications used in this study are available from the ECOD (http://prodata.swmed.edu/ecod/),

SCOP (https://scop.mrc-lmb.cam.ac.uk), and Pfam (https://www.ebi.ac.uk/interpro/) databases. Source data are provided with this paper.

## Code availability

Code used to generate the results reported in this manuscript is available at: https://doi.org/10.5281/zenodo.7837636 and https://github.com/ncbi/FixJ_KdpE.

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

## Acknowledgements
We thank Carolyn Ott for helpful discussions and Loren Looger for critically reading this manuscript. This work utilized the NIH HPS Biowulf cluster (http://hpc.nih.gov). It was supported in part by funding from the Intramural Research Program of the National Library of Medicine, National Institutes of Health (LM202011, L.L.P.), the National Institute of General Medical Sciences, National Institutes of Health (GM118589 to L.S.-K.) and the W. M. Keck Foundation (L.S.-K.).

## Author contributions
Conceptualization: L.L.P. and L.S.K. Methodology: L.L.P., D.C., L.S.K., and S.S. Software: D.C., L.L.P., and S.S. Investigation: L.L.P., D.C., L.S.K., and S.S. Data Curation: S.S., D.C., and L.L.P. Visualization: L.L.P., D.C., and S.S. Writing – original draft: L.L.P., D.C., and S.S. Writing – review & editing: L.L.P., L.S.K., D.C., and S.S. Supervision: L.L.P. and L.S.K. Project administration: L.L.P. Funding acquisition: L.L.P. and L.S.K.

## Competing interests
The authors declare no competing interests.
