## [Peer Review File · Nature Communications]

Identification of a covert evolutionary pathway between two protein foldsREVIEWER COMMENTS

Reviewer #1 (Remarks to the Author):

- What are the noteworthy results?

The paper examines the evolution of multidomain, bacterial response proteins. Some of these regulator proteins have a C-terminal, DNA binding domain with a helix-turn-helix motif while others have a C-terminal domain with a mixed alpha-beta structure called a winged helix.

The two versions of the C-terminal domain (CTD) do not have discernable homology, but might have evolved from a common ancestor, nonetheless.

The paper uses statistical methods to examine structure and sequence databases to determine if the two CTD versions evolved from a common ancestor via point mutations, as opposed to evolving independently, as DNA binding cassettes, which were later recombined with N-terminal portions.

When entire bacterial response proteins are aligned using homologous N-terminal regions to establish a register, subtle similarity among C-terminal domains suggests common ancestry.

This paper shows evidence that continuous pathways can be inferred from the current sequence and structure data bases.

- Will the work be of significance to the field and related fields?

Continuous evolutionary pathways between two proteins of fundamentally different structures have been proven previously based on the ability to create these pathways in designed proteins.

While the previous design work proves that such continuous pathways exist, this paper is a solid attempt to develop methods to prove that continuous pathways between different folds were used in natural evolution.

These methods lay a groundwork that will likely stimulate efforts to infer pathways between less similar folds. Thus, while conclusions are not startling, it is important to have evidence for the natural mechanism of fold switching in evolution.

- Does the work support the conclusions and claims?

Yes.

- Are there any flaws in the data analysis, interpretation and conclusions? Do these prohibit publication or require revision?

No.

- Is the methodology sound? Does the work meet the expected standards in your field?

Yes.

- Is there enough detail provided in the methods for the work to be reproduced?

Yes.

Reviewer #2 (Remarks to the Author):

Summary

The manuscript details a comparison between two bacterial DNA binding receptors that differ from each other in their mode of interaction with the DNA. Evidence from sequence and structure comparisons, using various tools, strongly indicate secondary structural changes that result in evolving from one DNA binding mode to another. Evolutionary reconstruction suggests evolutionary pathways between the two DNA binding modes.

Opinion

I am split regarding the suitability of the study for publication in Nature Commun. On the one hand, this beautifully written manuscript, reveals an interesting evolutionary story. On the other hand, that secondary structure often changes in evolution is well established. Furthermore, the manuscript says that the approach can be applied to other cases, but for that to happen one needs to know where to look for changes. That is, the manuscript does not provide a way to detect changes in secondary structure, but only to characterize them when there is hint.

Specific points:

- 1) Authors acknowledge the advantage of HMM-based searches and yet use mainly BLAST and PSI-BLAST, which are significantly inferior. They refer to equivalent studies that search for non-trivial similarities. While being aware of these, they nevertheless opt to much less accurate and sensitive BLAST-based comparisons. Why?
- 2) The first part of the study is based on a very limited search over the PDB, which contains a small (and not necessarily faithfully representative) fraction of known proteins. I would consider deleting this part.
- 3) Figure 3. Bootstrap values are missing. And CLUSTAL is not the most suitable tool for constructing MSA. MUSCLE and PRANK are usually better. Why was it used here? And elsewhere?
- 4) Most importantly, the manuscript refers to the helix-turn-helix vs. winged-helix as different folds. Are they really? For example, how do ECOD, Pfam and other domains database refer to them? Are they not known already to be linked? Anyway, they appear more like "fragments" or "themes", i.e., smaller than domains. I think that this point should be made clear.
- 5) With that, the study looks like a specific case of the broader studies that my colleagues and I conducted about protein segments that are shared between domains of seemingly different evolutionary origins, i.e., "bridging themes" (references 54 and 56). We showed that structure is conserved only in about half of the bridging themes, and pointed out cases with alpha-to-beta alterations. It should be made clear what the current study adds above and beyond these studies.
- 6) To further consolidate the proposed evolutionary scenario the authors may consider mapping the helix-turn-helix vs. winged-helix into a tree of life. This may also allow to decipher the direction of change. That is, helix-turn-helix \diamond winged-helix or the other way around. Or perhaps, back and forth.

Reviewer #3 (Remarks to the Author):

Report on Nat Comm Ms. by Chakravarty et al., Ms. No.: NCOMMS-22-50589

This is a very interesting and remarkably innovative study that leverages multiple bioinformatics techniques to address important questions of structural transformation in protein evolution. The results are impactful, suitable for the general audience of Nat Comm, and thus should be published in Nat Comm. Nonetheless, as noted below, the manuscript should be revised before publication to improve on the presentation of the authors' investigative logic, discussion of future prospects, and to draw better connections with highly relevant prior works so as to place the authors' effort in a more comprehensive scientific context.

1. Beginning on line 180 (page 5 of the manuscript), the authors describe a procedure using 3 rounds of PSI-BLAST. Was this done for the full-length proteins (not just their CTDs)? Earlier in the discussion (starting on line 153), it is reported that applying BLAST to full-length FixJ and KdpE did

not indicate their evolutionary relationship. For the general readership of Nat Comm, it will be useful to clarify the difference in the two approaches and the basic mathematical /biophysical reasons why the 3-round algorithm is more sensitive to the "hidden" evolutionary relationship between FixJ and KdpE.

2. The authors have effectively developed a bioinformatics protocol for discovering (putative) evolutionary pathways between different protein structures in the PDB. Do the authors expect their method to work in general (for other structural transformations)? If so, it will be useful to spell out the protocol in a step-by-step manner in the Discussion/Conclusion part of the manuscript.

3. The authors have predicted bridge (or "switch") sequences that may significantly populate both the FixJ and KdpE folds simultaneously. In this regard, it will be extremely interesting to conduct (wet-lab) experiments on these putative bridge/switch sequences. Although such experiments are beyond the scope of the present work, it will be useful to discuss the prospects. For example, early experiments on the arc repressor (Sauer group at MIT) indicates that a bridge sequence populates both the alpha and the beta dimers [Cordes et al., Science 284:325-7 (1999), Nat Struct Biol 7:1129-32 (2000)]. These works should be cited in this manuscript and included in the discussion.

4. Moreover, an early experimental study of the reconstructed common ancestor of red- and green-fluorescent proteins in coral indicates that the reconstructed ancestor can emit both red and green light [Ugalde et al., Science 305:1433 (2004)]. This intriguing example should be mentioned as well.

5. Evolution of new protein structure most likely involves gene duplication. The evolutionary dynamics of such processes in sequence space have been modeled [Sikosek et al., Proc Natl Acad Sci USA 109:14888-93 (2012)] and reviewed in detail [Sikosek & Chan, J Royal Soc Interface 11:20140419 (2014) <https://royalsocietypublishing.org/doi/full/10.1098/rsif.2014.0419>]. It will be useful to include this perspective in the manuscript. See in particular Figs.4, 8 and 9 of the 2014 review. There are additional examples of structural transformations in Fig.4 of this reference that the authors may wish to cite in their introductory discussion as well.

6. The GA-GB structural transformation (refs.19-20 to the Orban and Bryan groups) has been rationalized biophysically by a "hybrid" molecular dynamics model [Sikosek et al., PLoS Comput Biol 12:e1004960 (2016)]. The insights from this work are relevant to the authors' effort. Not only is this 2016 study useful for expanding the context of the authors' investigation, the approach developed in this reference can be useful for testing [as a more efficient computational method complementary to experiments] whether the authors' predicted evolutionary path shows an expected gradual decrease in stability of the original structure and a concomitant increase in stability of the target structure.

Once the suggested revisions described above are made, the manuscript should be reconsidered favorably for publication.

RESPONSE TO REVIEWERS' COMMENTS

We thank the reviewers for their positive comments and constructive suggestions. We have made changes throughout the manuscript to address these comments and improve clarity. Answers to specific questions are noted below in bold text.

Reviewer #1 (Remarks to the Author):

We thank Reviewer 1 for their positive comments and accurate summaries of our results.

- What are the noteworthy results?

The paper examines the evolution of multidomain, bacterial response proteins. Some of these regulator proteins have a C-terminal, DNA binding domain with a helix-turn-helix motif while others have a C-terminal domain with a mixed alpha-beta structure called a winged helix.

The two versions of the C-terminal domain (CTD) do not have discernable homology, but might have evolved from a common ancestor, nonetheless.

The paper uses statistical methods to examine structure and sequence databases to determine if the two CTD versions evolved from a common ancestor via point mutations, as opposed to evolving independently, as DNA binding cassettes, which were later recombined with N-terminal portions.

When entire bacterial response proteins are aligned using homologous N-terminal regions to establish a register, subtle similarity among C-terminal domains suggests common ancestry.

This paper shows evidence that continuous pathways can be inferred from the current sequence and structure data bases.

- Will the work be of significance to the field and related fields?

Continuous evolutionary pathways between two proteins of fundamentally different structures have been proven previously based on the ability to create these pathways in designed proteins.

While the previous design work proves that such continuous pathways exist, this paper is a solid attempt to develop methods to prove that continuous pathways between different folds were used in natural evolution.

These methods lay a groundwork that will likely stimulate efforts to infer pathways between less similar folds. Thus, while conclusions are not startling, it is important to have evidence for the natural mechanism of fold switching in evolution.

- Does the work support the conclusions and claims?

Yes.

- Are there any flaws in the data analysis, interpretation and conclusions? Do these prohibit publication or require revision?

No.

- Is the methodology sound? Does the work meet the expected standards in your field?

Yes.

- Is there enough detail provided in the methods for the work to be reproduced?

Yes.

Reviewer #2 (Remarks to the Author):

Summary

The manuscript details a comparison between two bacterial DNA binding receptors that differ from each other in their mode of interaction with the DNA. Evidence from sequence and structure comparisons, using various tools, strongly indicate secondary structural changes that result in evolving from one DNA binding mode to another. Evolutionary reconstruction suggests evolutionary pathways between the two DNA binding modes.

Opinion

I am split regarding the suitability of the study for publication in Nature Commun. On the one hand, this beautifully written manuscript, reveals an interesting evolutionary story. On the other hand, that secondary structure often changes in

evolution is well established. Furthermore, the manuscript says that the approach can be applied to other cases, but for that to happen one needs to know where to look for changes. That is, the manuscript does not provide a way to detect changes in secondary structure, but only to characterize them when there is hint.

We thank the Reviewer for their positive comments.

In response to this comment and that of Reviewers 3, we have included both text and a figure in the Discussion detailing how the approach described in this manuscript could be applied in other contexts.

Specific points:

1) Authors acknowledge the advantage of HMM-based searches and yet use mainly BLAST and PSI-BLAST, which are significantly inferior. They refer to equivalent studies that search for non-trivial similarities. While being aware of these, they nevertheless opt to much less accurate and sensitive BLAST-based comparisons. Why?

BLAST and PSI-BLAST were used for initial searches (Figure 1) because they are faster than HMMER, especially for a sequence set of this size; once we narrowed down the sequence space of interest, we used jackhmmer to validate our findings (Figures 2 and S1). This section of the text has been substantially revised to clarify this point.

2) The first part of the study is based on a very limited search over the PDB, which contains a small (and not necessarily faithfully representative) fraction of known proteins. I would consider deleting this part.

This step was critical to identify evolved fold switching: It allowed us to identify a pair of homologous proteins with different experimentally determined folds. Indeed, it directly pertains to Reviewer 2 and 3's comments about general applicability, as it was the key first step that can be used to identify such relationships in other proteins. Thus, we choose to keep this section in the manuscript; it has been substantially revised to improve the clarity of our rationale.

3) Figure 3. Bootstrap values are missing. And CLUSTAL is not the most suitable tool for constructing MSA. MUSCLE and PRANK are usually better. Why was it used here? And elsewhere?

We used both CLUSTAL Omega and MUSCLE alignments for this work. The similar results from two different alignment algorithms enhances confidence in our conclusions. In addition, our studies with very large sequence sets used CLUSTAL for computational efficiency. Results from MUSCLE, including the alignment, are included in the Supplement. In brief, highly significant bootstrap values were reported in Figure S5b and a transitive homology path between HTH and wH folds from the MUSCLE alignment was included in Figure S3. We have now added the full maximum likelihood tree from the MUSCLE alignment (Figure S6). Consistent with the CLUSTAL alignment, this tree suggests that the bridge sequences adjoin HTH and wH folds. Thus, in this case, the CLUSTAL alignment was suitable to give us highly consistent results, including with the maximum likelihood tree generated from the MUSCLE alignment.

4) Most importantly, the manuscript refers to the helix-turn-helix vs. winged-helix as different folds. Are they really? For example, how do ECOD, Pfam and other domains database refer to them? Are they not known already to be linked? Anyway, they appear more like “fragments” or “themes”, i.e., smaller than domains. I think that this point should be made clear.

Pfam, ECOD, and SCOP do recognize helix-turn-helix and winged helix as different folds. The classifications are as follows:

- Pfam: PF00010 (helix-turn-helix), PF02319 (winged helix)
- ECOD puts them in different T-groups (tetrahelical HTH and winged)
- SCOP: HTH: 8034563 (Superfamily C-terminal effector domain of the bipartite response regulators) Winged helix: 8075578 (Superfamily: PhoB-like)

We have added this information to the revised manuscript.

Common ancestry between HTH and wH has been proposed previously, at least as far back as Aravind Iyer, whose work we cited. This common ancestry was inferred from the trihelical bundle conserved between the two folds.

Nevertheless, to our knowledge the origin of N-terminal beta sheet and the C-terminal wing in winged helices has not been investigated. Indeed, without the extensive information recently deposited into sequence and structure databases, it could not have been identified. The purpose of this paper was to investigate their evolutionary mechanism; we found stepwise mutation to be the most likely mechanism for evolution of the C-terminal wing. As noted by Reviewer 1, evolutionary pathways between different folds through stepwise mutation have

been demonstrated in engineered systems but had not been discovered in nature prior to this paper.

Further, we now explain in detail how this switch differs from “themes” in the discussion. We summarize those differences in response to the question below.

5) With that, the study looks like a specific case of the broader studies that my colleagues and I conducted about protein segments that are shared between domains of seemingly different evolutionary origins, i.e., “bridging themes” (references 54 and 56). We showed that structure is conserved only in about half of the bridging themes, and pointed out cases with alpha-to-beta alterations. It should be made clear what the current study adds above and beyond these studies.

This study differs from “bridging themes” in several notable ways, and we have added these differences to the Discussion:

- (1) “Bridging themes” are defined as “homologous protein segments found in different sequential and structural contexts.” (Kolodny, et al. 2021, Mol. Biol. Evol.). Thus, the isolated sequences of bridging themes have discernable homology *without* the context of the rest of the protein. In contrast, the isolated sequences of HTH and wH fragments described in this work lack discernible homology without the context of the rest of the protein. Instead, to find their relationship, we needed to use full sequences. As Reviewer 1 noted, this is one of the noteworthy results of our paper.**
- (2) The evolutionary mechanisms for generating new folds differ between bridging themes and the fold switch that we have described. Bridging themes have been defined as conserved protein fragments that recombine with non-homologous segments of protein structure to form distinct domains (different ECOD X-groups). The fold switch reported here supports fold divergence by stepwise mutation. Restating, fragment recombination and stepwise mutation within a conserved protein fold are distinct modes by which new folds can evolve.**
- (3) Bridging themes (or their homologs) can adopt different structures in different protein contexts, much like chameleon sequences studied by Minor & Kim and Nick Grishin’s group. In contrast, the homologous sequences reported here have different structures in homologous protein contexts (*i.e.*, both the wH and HTH are C-terminal to a conserved trihelical helix-turn-helix fold).**

6) To further consolidate the proposed evolutionary scenario the authors may consider mapping the helix-turn-helix vs. winged-helix into a tree of life. This may also allow to decipher the direction of change. That is, helix-turn-helix \diamond winged-helix or the other way around. Or perhaps, back and forth.

We thank the Reviewer for their suggestion but expect that it will likely not yield a clear answer. Each bacterial strain has many response regulators with different sequences that may have evolved or been horizontally transferred at different times. Thus, it would be difficult to map these many sequences to an evolutionary tree with confidence. However, our ancestral sequence reconstructions and previous work from Aravind Iyer suggests that helix-turn-helix came before winged helix.

Reviewer #3 (Remarks to the Author):

Report on Nat Comm Ms. by Chakravarty et al., Ms. No.: NCOMMS-22-50589

This is a very interesting and remarkably innovative study that leverages multiple bioinformatics techniques to address important questions of structural transformation in protein evolution. The results are impactful, suitable for the general audience of Nat Comm, and thus should be published in Nat Comm. Nonetheless, as noted below, the manuscript should be revised before publication to improve on the presentation of the authors' investigative logic, discussion of future prospects, and to draw better connections with highly relevant prior works so as to place the authors' effort in a more comprehensive scientific context.

We thank the Reviewer for their positive comments.

1. Beginning on line 180 (page 5 of the manuscript), the authors describe a procedure using 3 rounds of PSI-BLAST. Was this done for the full-length proteins (not just their CTDs)? Earlier in the discussion (starting on line 153), it is reported that applying BLAST to full-length FixJ and KdpE did not indicate their evolutionary relationship. For the general readership of Nat Comm, it will be useful to clarify the difference in the two approaches and the basic mathematical /biophysical reasons why the 3-round algorithm is more sensitive to the "hidden" evolutionary relationship between FixJ and KdpE.

We ran PSI-BLAST on both the CTDs and the full-length response regulators. We have revised the text to better highlight the importance of this step. For example, in the middle of Page 5, we report the results from the CTD searches:

PSI-BLAST searches of the PDB using the sequences of isolated CTDs from either FixJ_{PDB} or KdpE_{PDB} as queries only identified sequences from the same fold families (HTH₄ or wH). Sequences encoding the alternative structure were not identified.

At the bottom of the page, we report the results from the full-length searches:

“...we next used *full-length* FixJ_{PDB} to query the PDB with 3 rounds of PSI-BLAST³⁹... **This alignment approach also** shifted the alignment registers of the CTDs, **so that** 97% of the FixJ_{PDB} sequence aligned with KdpE_{PDB} with an e-value of 6×10^{-39} (Figure 1b, right)...

...A reciprocal, three-round PSI-BLAST search using the full-length KdpE_{PDB} sequence as query aligned 90% of this protein with FixJ_{PDB}, with an e-value of 10^{-29} .”

We now explain differences between BLAST and PSI-BLAST as follows:

“Unlike the faster BLAST algorithm (which identifies matches using pairwise identities between the query sequence and entries in a sequence database), PSI-BLAST searches for sequences that match conservation patterns within a set of homologous sequences used to generate a position-specific scoring matrix. This matrix stores scores for substituting one amino acid for another in each sequence position and is updated after each PSI-BLAST iteration if new sequences are hit in the search. As such, PSI-BLAST identifies hidden conservation patterns characteristic to a given protein family that cannot be detected by BLAST.”

2. The authors have effectively developed a bioinformatics protocol for discovering (putative) evolutionary pathways between different protein structures in the PDB. Do the authors expect their method to work in general (for other structural transformations)? If so, it will be useful to spell out the protocol in a step-by-step manner in the Discussion/Conclusion part of the manuscript.

We thank the Reviewer for this excellent point. We have included a step-by-step protocol (with figure) in the expanded Discussion so that our approach can be applied more broadly.

3. The authors have predicted bridge (or “switch”) sequences that may significantly populate both the FixJ and KdpE folds simultaneously. In this regard, it will be extremely interesting to conduct (wet-lab) experiments on these putative bridge/switch sequences. Although such experiments are beyond the scope of the

present work, it will be useful to discuss the prospects. For example, early experiments on the arc repressor (Sauer group at MIT) indicates that a bridge sequence populates both the alpha and the beta dimers [Cordes et al., Science 284:325-7 (1999), Nat Struct Biol 7:1129-32 (2000)]. These works should be cited in this manuscript and included in the discussion.

We thank the Reviewer for this point and have included a new paragraph discussing this in the discussion. Regarding the work from the Sauer group, we added the following with citations:

“As previous work has shown, structural interconversion can be observed with nuclear magnetic resonance (NMR) spectroscopy. Indeed, NMR studies of the Arc repressor and XCL1 identified a handful of key mutations that switch protein folds. Accordingly, it would be interesting to experimentally identify minimal mutational pathways that switch HTH₄ sequences to wH and vice versa.”

4. Moreover, an early experimental study of the reconstructed common ancestor of red- and green-fluorescent proteins in coral indicates that the reconstructed ancestor can emit both red and green light [Ugalde et al., Science 305:1433 (2004)]. This intriguing example should be mentioned as well.

Added as follows, with references included in the main text:

“Although outside the scope of this study, experimental testing of the reported bridge sequences and reconstructed ancestors may reveal mechanistic details of the transition from HTH₄ to wH. Whether any of these sequences populate both folds – as has been observed for other fold-switching proteins– would be of particular interest. For the reconstructed ancestors, structural interconversion would be analogous to functional studies of reconstructed ancestors of green and red fluorescent proteins that emit both green and red light...”

5. Evolution of new protein structure most likely involves gene duplication. The evolutionary dynamics of such processes in sequence space have been modeled [Sikosek et al., Proc Natl Acad Sci USA 109:14888-93 (2012)] and reviewed in detail [Sikosek & Chan, J Royal Soc Interface 11:20140419 (2014) <https://royalsocietypublishing.org/doi/full/10.1098/rsif.2014.0419>]. It will be useful to include this perspective in the manuscript. See in particular Figs.4, 8 and 9 of the 2014 review. There are additional examples of structural transformations in Fig.4 of this reference that the authors may wish to cite in their introductory discussion as well.

We added these studies as follows, with references included in the main text:

“Biophysically based computational approaches may also provide insights into the mechanism and evolution of response regulator proteins with HTH₄ and wH domains. Such studies successfully predicted fold transitions between engineered protein G variants with high levels of sequence identity but different folds. Other biophysical models or hybrid theoretical-experimental approaches can be used to infer the roles of point mutations, multifunctionality, selective pressure, and epistasis in protein evolution.”

6. The GA-GB structural transformation (refs.19-20 to the Orban and Bryan groups) has been rationalize biophysically by a “hybrid” molecular dynamics model [Sikosek et al., PLoS Comput Biol 12:e1004960 (2016)]. The insights from this work are relevant to the authors’ effort. Not only is this 2016 study useful for expanding the context of the authors’ investigation, the approach developed in this reference can be useful for testing [as a more efficient computational method complementary to experiments] whether the authors’ predicted evolutionary path shows an expected gradual decrease in stability of the original structure and a concomitant increase in stability of the target structure.

We thank the reviewer for noting this point. We have added the following (with references included) in the main text:

“Such studies successfully predicted fold transitions between engineered protein G variants with high levels of sequence identity but different folds.”

Once the suggested revisions described above are made, the manuscript should be reconsidered favorably for publication.

REVIEWERS' COMMENTS

Reviewer #2 (Remarks to the Author):

The authors have addressed well all issues raised and the manuscript can be published. Cheers! Just to clarify, indeed, by construction, "bridging themes" have to present in (at least) two different ECOD X groups, and maybe half of these were found in different contexts. However, they were detected based on "themes", themselves detected based purely on sequence similarity. It may well be that the themes, which can be found among homologues, can nevertheless manifest different structures. We didn't check. Maybe we should.

Reviewer #3 (Remarks to the Author):

The authors have adequately addressed my previous concerns and suggestions. The description of the methods and their rationale as well as the discussion of related works are now much improved. The addition of the step-by-step guide (Fig.6 in the revised manuscript and related discussion) is particularly useful. Accordingly, I recommend publication of the revised manuscript in Nat Comm.